# Hypoxia and Tissue Regeneration: Adaptive Mechanisms and Therapeutic Opportunities

**DOI:** 10.3390/ijms26199272

**Published:** 2025-09-23

**Authors:** Isabel Cristina Vásquez Vélez, Carlos Mario Charris Domínguez, María José Fernández Sánchez, Zayra Viviana Garavito-Aguilar

**Affiliations:** 1Departamento de Ciencias Biológicas, Facultad de Ciencias, Universidad de los Andes, Bogota 111711, Colombiacm.charris@uniandes.edu.co (C.M.C.D.); 2Departamento de Ciencias Fisiológicas, Facultad de Medicina, Pontificia Universidad Javeriana, Bogota 110411, Colombia; m.fernandez@javeriana.edu.co; 3Hospital San Ignacio, Unidad de Neumología, Bogota 111311, Colombia

**Keywords:** tissue regeneration, hypoxia, HIF signaling, cellular adaptation, comparative models, oxygen homeostasis

## Abstract

Reduced oxygen availability, or hypoxia, is an environmental stress factor that modulates cellular and systemic functions. It plays a significant role in both physiological and pathological conditions, including tissue regeneration, where it influences angiogenesis, metabolic adaptation, inflammation, and stem cell activity. Hypoxia-inducible factors (HIFs) orchestrate these responses by activating genes that promote survival and repair, although HIF-independent mechanisms, particularly those related to mitochondrial function, are also involved. Depending on its duration and severity, hypoxia may exert either beneficial or harmful effects, ranging from enhanced regeneration to fibrosis or maladaptive remodeling. This review explores the systemic and cellular effects of acute, chronic, intermittent, and preconditioning hypoxia in the context of tissue regeneration. Hypoxia-driven responses are examined across tissues, organs, and complex structures, including the heart, muscle, bone, vascular structures, nervous tissue, and appendages such as tails. We analyze findings from animal models and in vitro studies, followed by biomedical and pharmacological strategies designed to modulate hypoxia and their initial exploration in clinical settings. These strategies involve regulatory molecules, signaling pathways, and microRNA activity, which are investigated across species with diverse regenerative capacities to identify mechanisms that may be conserved or divergent among taxa. Lastly, we emphasize the need to standardize hypoxic conditions to improve reproducibility and highlight their therapeutic potential when precisely controlled.

## 1. Introduction

Hypoxia denotes a reduction in oxygen availability to humans and other animals. In general, oxygen partial pressures (pO2) below 24–30 mmHg for tissue and less than 10 mmHg for the intracellular environment are considered hypoxic, although specific thresholds may vary depending on the tissue [1,2,3]. Due to its key role in cellular energy production, a reduced oxygen supply impairs function and threatens viability in both cells and metazoan organisms. It is not surprising that hypoxia happens in both pathological and physiological contexts, particularly in mammals, including humans. Hypoxic conditions may arise acutely (seconds to hours), chronically (from days, weeks, or even permanently), or intermittently (with transient exposures followed by normoxia). The decrease in oxygen availability can be systemic and naturally occurring under hypobaric conditions at high altitudes, where the atmospheric partial pressure of oxygen decreases from approximately 159 mmHg at sea level to about 108 mmHg at 2500 m, and nearly 75 mmHg at 5000 m [4]. According to the International Society for Mountain Medicine classification, these elevations correspond to high (2500–3500 m), very high (3500–5500 m), and extreme altitude (>5500 m), respectively. This progressive decline contributes to reduced alveolar and tissue oxygenation. Considering that more than 500 million people live above 1500 m and approximately 81.6 million reside above 2500 m, the study of environmental hypoxia in these contexts becomes highly relevant [4]. Beyond environmental exposure, hypoxia also arises in pathological contexts such as ischemia, which involves a reduction or interruption of blood flow to tissues, causes a marked drop in local pO_2_ that may fall below 20 mmHg in the affected areas, and plays a fundamental role in the pathogenesis of various cardiovascular diseases [5,6].

Clarifying what constitutes hypoxic conditions across biological and experimental systems is essential for interpreting physiological and pathological responses to reduced oxygen availability. While general thresholds have been proposed, specific definitions may vary depending on the species, cell type, and experimental methodology used in each case. In humans, arterial hypoxemia is commonly defined as a pO_2_ below 100 mmHg (approximately 13–14% O_2_), although physiological tissue and intracellular values are typically much lower [7]. In experimental models, particularly in rodents, hypoxia is often induced using the Fraction of inspired oxygen (FiO_2_) values between 5% and 6%, corresponding to pO_2_ levels of approximately 35–43 mmHg [8]. In in vitro systems, oxygen concentrations between 1% and 3% (pO_2_ ≈ 7–21 mmHg) are generally classified as moderate hypoxia, while values between 0.1% and 0.5% O_2_ (≈3.5 mmHg pO_2_) are considered severe. Significantly, conventional “normoxic” cell culture conditions (~20% O_2_, ≈approximately 140 mmHg pO_2_) exceed physiological oxygen levels, which in most tissues range from 2% to 9% O_2_ (≈approximately 14–70 mmHg) [9]. Recognizing these distinctions is crucial when evaluating hypoxia-driven cellular processes across experimental and physiological contexts.

Therefore, the adaptive mechanisms that enable survival and cellular recovery under low oxygenation have motivated active research in multiple biomedical areas. Many of these studies focus on hypoxia-inducible factor (HIF), which is the primary regulator of the cellular response to hypoxia. HIF leads the cell toward changes in metabolism to obtain energy more efficiently. HIF also promotes adaptive responses by increasing angiogenesis to improve perfusion for cells, activating protection against oxidative stress, and supporting survival pathways [10]. All these responses can profoundly influence the regenerative capacity of tissues. While these adjustments are mostly considered beneficial, in cases when hypoxia is severe or prolonged, it can lead to maladaptation. An example of this occurs in chronic mountain sickness, where insufficient cardiopulmonary adaptation at high altitudes leads to sustained hypoxemia and its complications. Another example is observed in cancer, where tumor cells in hypoxic microenvironments activate mechanisms that enhance survival and malignancy [11,12].

Thus, we analyze how organisms, ranging from single cells to humans, respond to various forms of hypoxia and how these responses influence regenerative processes with significant biomedical implications. This review addresses these questions by first describing the systemic and cellular responses to acute, chronic, and intermittent hypoxia, which are centrally mediated by HIFs, as well as by some HIF-independent regulators, leading to either physiological adaptation or pathological outcomes. We then examine how these hypoxia-driven responses influence tissue regeneration across various organs and systems, including the heart, muscle, bone, neural tissue, vascular structures, and complex structures such as appendages. Finally, we highlight biomedical and clinical approaches that aim to harness hypoxia for regenerative medicine, including preconditioned stem cells, extracellular vesicles, hydrogels, and other bioengineering strategies, as well as emerging clinical studies in this field. This review underscores both the therapeutic potential of hypoxia and the urgent need to standardize methodologies when applying hypoxic conditions, given their context-dependent and potentially harmful effects.

## 2. Hypoxia Types and Systemic Adaptation

The nature of hypoxic exposure, whether brief, prolonged, or cyclical, influences both the immediate response and the long-term adaptations. All these critical stimuli generate adaptive responses in cells, tissues, and organisms, affecting both survival and regenerative potential in both homeostasis and disease [13]. Therefore, the duration: acute, defined as minutes to hours; chronic, as days to years; and intermittent, as alternating periods of hypoxia and normoxia, is a key determinant. Furthermore, the severity and level of organismal adaptation to these conditions are key determinants of the type of response and its ultimate impact on the individual [14].

These responses can be observed at the cellular, systemic, and whole-organism levels. The central regulator and driver of these changes at the cellular level is HIF. This factor modulates the expression of numerous genes essential for adaptation, and its regulatory mechanisms will be explained in detail in the following section. Under HIF activity, profound changes occur within the cell. For example, it stimulates a more efficient metabolism based on anaerobic glycolysis, which leads to a temporary increase in lactate, a reduction in mitochondrial oxygen consumption, and a decrease in cellular dependence on this gas [15,16]. The reduction in mitochondrial activity limits the generation of free radicals and oxidative stress, especially during reoxygenation [16].

Additionally, HIF promotes angiogenesis, which ultimately increases tissue perfusion and improves oxygenation [17]. Many of these adaptive strategies are conserved in vertebrates and invertebrates, demonstrating their evolutionary importance. These mechanisms, preserved through evolution, form the basis for immediate responses to sudden hypoxic exposure.

In contexts of acute hypoxia in humans, the initial response is to maximize the capture of available environmental oxygen. This oxygen uptake is achieved by activating chemoreceptors, such as the carotid body, resulting in increased depth and frequency of breathing. This change is accompanied by an increase in pulmonary vascular resistance, which aims to improve the ventilation-perfusion ratio and thus gas exchange [17]. Although ventilatory responses are often attenuated, changes in pulmonary circulation are variable. These responses can be sustained or even intensified, depending on the severity of hypoxia and individual adaptation to these conditions. Sometimes, this can lead to pulmonary hypertension. Early activation of angiogenesis and prioritization of anaerobic glycolysis are also observed [17,18].

The immune response also changes during acute hypoxia, with an increase in proinflammatory signaling and innate immune activity. In the first scenario of acute hypoxia, initial immune activation has been described in human subjects exposed to high-altitude hypoxia [19]. Mechanistic evidence from animal models and in vitro systems supports these observations [16,20]. The immediate responses are induced by HIF-1α (hypoxia-inducible factor 1 alpha), one of the most critical HIF isoforms, and its influence on the NF-κB (Nuclear Factor κB) pathway, which is involved in inflammatory and immune responses [16,19,20]. However, this initial context is counterbalanced by the activity of NRF2 (Nuclear factor erythroid 2-related factor 2), a transcription factor involved, among other things, in cellular defense against oxidative stress [19]. Together, these responses, under acute hypoxia, aim to increase protection and tissue recovery in the face of potential exposure to Damage-Associated Molecular Patterns (DAMPs), molecules released by injured cells, oxidative stress, and reperfusion-reoxygenation injury [16,19,20]. The second scenario is chronic hypoxia. In this case, depending on the duration of the stimulus, ventilatory changes and sympathetic effects on the cardiovascular system are attenuated. This moderation also occurs in immune activity that shifts toward an anti-inflammatory predominance, promoting resolution and tissue remodeling processes. This adjustment is mainly mediated by HIF-2α (hypoxia-inducible factor 2 alpha), a key HIF isoform mostly related to long-term responses [20]. However, under chronic hypoxia, the most prominent changes are hematological, including increased erythropoiesis, higher hemoglobin concentrations, and, in some species, including humans, other mammals, and certain birds, a shift in hemoglobin oxygen affinity that enhances oxygen loading and delivery to tissues. This adaptation is particularly critical in species or populations that naturally inhabit chronically hypoxic environments [2,17,21,22,23].

These observations emphasize the diverse physiological responses to hypoxic stress across species. In humans, as in other vertebrates, these changes serve not only to ensure survival but also, in many cases, to facilitate successful establishment in such environments. Initially, under acute exposure, organisms attempt to acclimate through temporary and reversible physiological responses that subside once the stimulus finishes. However, when low oxygen levels persist, functional and structural modifications may help sustain homeostasis over more extended periods. In both humans and other species, such changes include phenotypic plasticity and, potentially, epigenetic reprogramming, although the latter is typically not stably inherited across generations [17]. In populations subjected to sustained hypoxia over many generations, stable genetic adaptations may arise through positive selection for advantageous alleles or negative selection against deleterious variants, a process documented in high-altitude human populations as well as in various animal models [21,23,24,25]. These genetic adaptations contribute to long-term evolutionary changes that support survival in chronically hypoxic environments.

Phenotypic and genotypic adaptations to chronic hypoxia have been observed across diverse vertebrate species. The relevant underlying genetic changes in different vertebrate species are listed in Table 1. Briefly, in teleost fish, such as the zebrafish (*Danio rerio*), hypoxia tolerance involves increased ventilation, enhanced cardiac output (i.e., the amount of blood the heart pumps per minute), and developmental plasticity that improves survival under low oxygen conditions. These phenotypes are supported by retained paralogs and regulatory adaptations in genes involved in oxygen sensing, erythropoiesis, and cellular metabolism [26]. In birds, several highland species exhibit enlarged lungs and pulmonary vascular remodeling that enhance oxygen diffusion [27,28]. In Andean house wrens (*Troglodytes aedon*), mutations in β-globin genes increase hemoglobin–oxygen affinity [22]. In bar-headed geese (*Anser indicus*), efficient high-altitude flight is enabled by hemoglobin adaptations, mitochondrial efficiency, and cardiovascular traits [2,21]. Among non-human mammals, species such as yaks (*Bos grunniens*), Tibetan sheep (*Ovis ammon hodgsoni*), and deer mice (*Peromyscus maniculatus*) exhibit a display of increased oxygen-carrying capacity, enhanced aerobic metabolism, and reduced pulmonary vascular resistance [29,30]. These characteristics are supported by genomic changes in genes under positive selection involved in oxygen sensing, ventilatory regulation, erythropoiesis, and mitochondrial metabolism [24]. Additional examples, including the North American pika (*Ochotona princeps*), are also included in Table 1.

Like the animal species previously described, specific human populations with ancestral residence at high altitudes have developed physiological, phenotypic, and genetic adaptations. Notable examples include populations inhabiting the highlands of Tibet, the Andes, and Ethiopia, all of which have evolved long-term responses to chronic hypoxia. Despite facing the same environmental challenge, these groups exhibit functionally convergent, yet genetically distinct adaptive strategies as detailed in Table 1. Tibetans, who have inhabited the Himalayas at an average elevation of over 3500 m for approximately 25,000 years, display elevated resting ventilation, low hemoglobin concentrations, and increased hemoglobin-oxygen affinity. They also show efficient oxygen utilization and enhanced circulation [31]. Genetic adaptations involve pathways regulating oxygen sensing, erythropoiesis, and mitochondrial metabolism [32,33]. Andean populations, residing above 3000 m for approximately 10,000 years, exhibit elevated hemoglobin levels and an enhanced capacity for oxygen transport [34]. Some also develop mild right ventricular hypertrophy and elevated pulmonary pressures, which may support gas exchange but predispose to chronic mountain sickness [18]. Genetic variants under selection affect cardiovascular regulation, erythropoiesis, and hypoxia tolerance [35,36]. Ethiopian highlanders, specifically the Amhara, who have lived above 2500 m for over 5000 years, maintain high arterial oxygen saturation without elevated hemoglobin levels and generally low pulmonary pressures [37]. Their genetic adaptations involve vascular function, hypoxia signaling, and regulation of oxidative stress [38].

These adaptations reflect independent evolutionary solutions that converge on similar physiological outcomes. Key traits include enhanced ventilation, increased erythropoiesis, improved oxygen affinity, and mitochondrial adaptations that protect vital organs from damage caused by hypoxia. A complete summary of species, genes, functions, and variants is provided in Table 1.

**Table 1 ijms-26-09272-t001:** Comparative genomic and phenotypic adaptations to chronic hypoxia in high-altitude vertebrates.

Species or Group	Genes or Identified Variants	Type of Variant	Biological Pathway or Function	Phenotype or Adaptive Trait	Adaptive Physiological Effect	Ref.
**Teleost fish**						
Wild-type Zebrafish (*Danio rerio*)	*hif-1aa*, *hif-1ab*, *hif-2aa*, *hif-2ab*, *hif-3aa*, *hif-3ab*	Coding; retained paralogs from teleost genome duplication	HIF signaling; oxygen sensing and erythropoiesis	Differential responses to hypoxia across tissues; *Hif-3* paralogs are required for erythropoiesis	Improved hypoxia tolerance; sub-functionalized oxygen response	[26]
Wild-type Zebrafish (*Danio rerio*)	*phd1*, *phd2*, *phd3*	Coding; genome duplication	Prolyl hydroxylation; regulation of HIF stability	Retention of multiple isoforms allows fine regulation	Enhanced capacity to modulate HIF degradation under variable oxygen levels	[26]
**Birds**						
Andean house wren (*Troglodytes aedon*)	β-globin (avian *HBB* ortholog)	Missense mutation in oxygen-binding site	Hemoglobin oxygen-binding affinity	Increased hemoglobin O_2_ affinity in high-altitude populations	Increased oxygen uptake and delivery to tissues in hypoxic environments	[22]
Andean ducks (*A. flavirostris, Spatula cyanoptera, A. georgica*)	*ND2*, *COI*, *ATP6* (mitochondrial genes)	Purifying selection on mitochondrial coding sequences	Oxidative phosphorylation efficiency	Enhanced ATP production under low oxygen availability	Efficient aerobic energy production in hypoxic high-altitude habitats	[28]
Bar-headed goose (*Anser indicus*)	α-globin, β-globin (*HBA, HBB* orthologs)	Amino acid substitutions with increased O_2_ affinity	Hemoglobin structure and gas transport	Efficient O_2_ transport in hypobaric conditions	Sustained aerobic flight at high altitude with low ambient oxygen pressure	[21]
**Non-human mammals**						
Deer mice (*Peromyscus maniculatu*)	*HIF-2* α *(EPAS1*), metabolic regulators	Functional SNP in knock-in models	Ventilatory drive and oxygen sensing	Reduced carotid body sensitivity; energy conservation	Energy conservation by reducing ventilation response to hypoxia	[29]
Yak (*Bos grunniens*)	*EPAS1, EGLN1, ADAM17*	Regulatory variants; differential gene expression	Oxygen sensing, angiogenesis, erythropoiesis	Low hematocrit, reduced pulmonary vascular resistance, blunted sympathetic response	Protection against hypoxia-induced pulmonary hypertension and efficient oxygen delivery	[30]
Tibetan sheep (*Ovis ammon hodgsoni*)	*EPAS1, EGLN1*	Functional polymorphisms; signals of positive selection	HIF pathway, ventilatory regulation	Enhanced ventilatory response, regulated hemoglobin concentration	Enhanced ventilation and stable blood oxygenation	[30]
American pika (*Ochotona princeps*)	Not specified (physiological traits documented)	Physiological adaptation; genetic basis not fully characterized	Pulmonary circulation and sympathetic regulation	Protection against pulmonary hypertension and cardiopulmonary stress	Resistance to pulmonary hypertension through reduced sympathetic tone	[24]
**Humans**						
Tibetans (*Homo sapiens*)	*EPAS1*	Introgressed non-coding SNPs	HIF regulation, erythropoiesis	Low hemoglobin concentration without hypoxia symptoms	Avoidance of chronic mountain sickness and erythrocytosis	[32]
Tibetans (*Homo sapiens*)	*EGLN1 (C127S*)	Coding SNP (missense mutation)	HIF hydroxylation, O_2_ homeostasis	Balanced erythropoiesis with reduced HIF activity	Fine-tuned erythropoiesis without excessive red blood cell production	[32]
Andeans (*Homo sapiens*)	*PRKAA1, EDNRA*	Regulatory polymorphisms	Cellular energy sensing and vascular regulation	Improved fetal outcomes and energy metabolism	Better fetal oxygen supply and vascular adaptation during pregnancy	[36]
Andeans (*Homo sapiens*)	*NOS2A, EGLN1*	Regulatory variants and non-synonymous SNPs	Nitric oxide synthesis, vascular function	Enhanced vasodilation and oxygen delivery	Increased NO-mediated vasodilation and oxygen delivery	[35]
Ethiopians (*Homo sapiens*)	*VAV3, BHLHE41*	Regulatory SNPs; signals of selection	Hematopoiesis, circadian regulation	Stable oxygen saturation at low Hb	Maintained oxygen saturation with lower hemoglobin levels	[37]
Ethiopians (*Homo sapiens*)	*CBARA1 (MICU1*)	Regulatory SNPs	Mitochondrial calcium uptake	Optimized mitochondrial metabolism in hypoxia	Improved mitochondrial calcium handling for energy efficiency	[38]

Table 1 summarizes phenotypic traits and genetic variants associated with adaptation to chronic hypoxia across selected high-altitude vertebrate species and human populations. Species are organized taxonomically (fish, birds, non-human mammals, and humans). For each case, the identified genes or variants, the type of genetic change (e.g., coding, regulatory, gene duplication), the affected functional pathway (e.g., oxygen sensing, erythropoiesis, metabolic regulation), the adaptive phenotype, and the proposed physiological advantage are described. Gene and protein names follow standard nomenclature conventions for each species. The abbreviation HIF stands for hypoxia-inducible factor; EPAS1 refers to endothelial PAS domain protein 1 (also known as HIF-2α); *EGLN1* encodes prolyl hydroxylase domain protein 2 (PHD2); EPO refers to erythropoietin; and PPARα is peroxisome proliferator-activated receptor alpha. Additional abbreviations include HBA (hemoglobin subunit alpha), HBB (hemoglobin subunit beta), COI (cytochrome c oxidase subunit I), ND2 (NADH dehydrogenase subunit 2), and ATP6 (ATP synthase subunit 6). In each case, adaptive traits are identified based on comparisons with either lowland conspecifics (e.g., in deer mice and house wrens), ancestral or reference genotypes (e.g., in humans), or normoxic control populations in experimental models (e.g., zebrafish). Where available, evidence of positive selection or functional impact is included to support the adaptive relevance of the genomic changes.

These phenotypic and genotypic adaptations to chronic hypoxia are associated with a protective effect against specific injuries or diseases. One example of this is seen in rats exposed to chronic pharmacological hypoxia, which showed greater and faster recovery after myocardial ischemia compared to controls [39]. However, there are contrasting findings in mice, also exposed to chronic hypoxia, which showed poor outcomes after an ischemic event [40]. These results from animal models under chronic hypoxia should be interpreted with caution, since experimental conditions and even the characteristics of the species used can significantly influence outcome variability.

The possible protective effect of chronic hypoxia has also been described in long-term high-altitude human residents. Reports from North American and European populations of permanent high-altitude residents (above 2500 m) show significant levels of cardioprotection, with lower incidences of atherosclerosis and myocardial infarction [41,42,43,44]. Positive effects and better recovery after ischemia and reperfusion have also been described in populations living above 2500 m [45,46]. These findings are particularly relevant, considering that approximately 140 million people worldwide live at altitudes of 2500 m or higher [47].

However, despite these apparent advantages, chronic hypoxia can also result in harmful effects when exposure is severe and prolonged. An observational study comparing permanent residents of La Rinconada (5100 m) and Puno (3800 m), both in Peru, revealed significant pathological changes in individuals living at the highest altitude. Residents at 5100 m, despite long-term exposure that would suggest better adaptation, displayed pathological cardiac remodeling, reduced cardiac ejection fraction, pulmonary hypertension, and a higher incidence of chronic mountain sickness (CMS). Among CMS patients, outcomes were markedly worse at extreme altitude [48]. These findings illustrate that intense and sustained hypoxia can override adaptive mechanisms, leading to maladaptive and potentially life-threatening outcomes. Chronic hypoxia has been associated with persistent inflammation, predominantly mediated by M1 macrophages, promoting fibrosis and endothelial dysfunction [49]. In patients under chronic hypoxic conditions, pulmonary fibrosis and alveolar barrier disruption have also been reported [50]. Moreover, excessive erythropoiesis, initially a compensatory response, can become maladaptive by increasing blood viscosity, raising thrombotic risk, overloading cardiac function, and impairing cerebral perfusion [34].

Additional consequences have been observed in long-term high-altitude ancestral residents, a population exposed to natural chronic hypoxia. Compared to lowland natives, they show reduced physical and cognitive performance [47], and individuals with insufficient adaptation are more prone to acute and chronic mountain sickness. Furthermore, chronic hypoxia, particularly in high-altitude populations, has been associated with worse outcomes in patients with pre-existing cardiorespiratory conditions [41,46].

These findings reveal the complex and context-dependent nature of chronic hypoxia, which can provide cardioprotective benefits under moderate conditions but lead to maladaptive effects under sustained and extreme exposure. Thus, the following related question arises: What happens to individuals who are exposed to hypoxia intermittently? This question is still under investigation from various perspectives. It introduces an important concept, “Hypoxic preconditioning,” which refers to a brief exposure to low oxygen that increases resistance to future stress. For example, in tumor cells, although not the focus of this review, exposure to hypoxia-reoxygenation cycles has been associated with advantages in proliferation, angiogenesis, metastasis, and survival [48]. One notable example comes from an animal model, in which mice preconditioned with short periods of hypoxia exhibit a protective effect against damage caused by reperfusion after ischemia [45].

Additionally, intermittent hypoxia (59 kPa, equivalent to ~4573 m) in Sprague-Dawley rats led to increased tolerance to extreme simulated hypoxia (40 kPa, corresponding to ~7620 m). This effect was partially explained by the associated activity of STAT3 (Signal Transducer and Activator of Transcription 3), RXR (Retinoid X Receptor), and Nrf2 (Nuclear factor erythroid 2-related factor 2), which are essential transcriptional factors for survival, repair, proliferation, and cellular protection against oxidative stress [51]. Therefore, short exposure or previous preadaptation to hypoxia appears to favor subsequent exposures, even those much more intense [15].

Clinical studies have reported the potential benefits of controlled intermittent hypoxia in humans, particularly in neurological and cardiovascular rehabilitation. Properly dosed protocols may enhance neuroplasticity and motor recovery after spinal cord injury or stroke, improve cardiovascular function and exercise tolerance, and optimize mitochondrial efficiency [52]. Complementing these benefits, intermittent hypoxia also modulates inflammatory responses through HIF- and NF-κB-dependent pathways, shifting the balance toward the resolution of inflammation, regulating immune cell activity, and stimulating cellular proliferation and migration that support tissue repair [52].

Together, these changes protect against further acute severe hypoxia or ischemic events [52]. These advantages are partly explained by the sustained action of HIF-1α, which is stimulated by VEGF (Vascular Endothelial Growth Factor) and EPO (Erythropoietin), promoting angiogenesis and improving hemoglobin concentrations and oxygen transport to tissues. HIF-1α also influences the activity of NF-κB and NRF2, which promote early immune responses, inflammation, and antioxidant protection [52]. Likewise, improved endothelial integrity has been reported after inducing ischemia in healthy patients previously preconditioned with hypoxia [53]. However, the positive or negative effects of intermittent hypoxia depend directly on the intensity, duration, and repetition of the exposure events [52,54].

While short and moderate intermittent hypoxia protocols have shown promising results in preconditioning, repeated exposure over extended periods can become detrimental. These effects are particularly evident in patients with obstructive sleep apnea (OSA), where chronic intermittent hypoxia leads to sustained activation of the HIF-1α/NF-κB axis, promoting inflammation, increased sympathetic activity, and systemic hypertension [55,56]. These pathophysiological changes observed in humans have been further explored in animal models, where chronic intermittent hypoxia has been associated with oxidative stress, persistent sympathetic overactivation, myocardial fibrosis, and vascular and renal sclerosis [55]. Thus, although intermittent hypoxia can enhance resilience under controlled conditions, its chronic form, as observed in OSA, poses significant health risks that must be carefully considered in therapeutic contexts.

Up to this point, the powerful effects of hypoxia, beneficial or detrimental, are evident. These depend on the context in which low oxygen availability occurs, its duration, severity, and the repetition of exposure events. Particularly, moderate chronic hypoxia and some acute intermittent hypoxia conditions have been shown to produce potentially beneficial effects. These processes involve, among others, immune responses, angiogenesis, cellular proliferation, protection against oxidative stress, and more efficient use of cellular energy resources. All these elements are relevant in the context of tissue repair and, potentially, regeneration. In the following section, we will examine the HIF-regulated cellular machinery that enables adaptation and coordinated responses to low oxygen availability.

## 3. Cellular Mechanisms Under Hypoxia

### 3.1. Cellular Mechanisms of Adaptation to Hypoxia Mediated by HIF

To understand how these systemic responses are coordinated, it is essential to examine the role of HIF in mediating cellular adaptation to hypoxia. The role of HIF as a critical regulator under low oxygen availability remained unknown until 1991, when it was uncovered by Semenza and his research group [57]. They discovered the RCGTG-3′ sequence in the *EPO* gene, which ultimately became a consensus sequence, later named HREs (Hypoxia Response Elements). This sequence is not exclusive to *EPO*, which encodes the EPO protein involved in erythrocyte production, responsible for transporting oxygen to tissues. However, it is present in more than 70 genes, enabling them to effectively interact with HIF, a transcription factor that promotes their expression. This activation allows cells and tissues to adapt and function more efficiently under low-oxygen conditions, promoting angiogenesis, metabolic adaptations, and cellular proliferation, among other effects [58].

This adaptive machinery is only activated when HIF induces the expression of its target genes. For this, heterodimerization and nuclear translocation of its two subunits are required: HIF-1β, also known as ARNT (aryl hydrocarbon receptor nuclear translocator), and HIF-α, which exists in three main isoforms: HIF-1α, HIF-2α, and HIF-3α. These two HIF subunits (α and β) belong to the basic helix-loop-helix PER-ARNT-SIM (bHLH-PAS) family of transcription factors [58,59]. Structurally, both subunits contain an N-terminal bHLH domain for DNA binding and PAS-A and PAS-B domains, which are required for dimerization. They also include transactivation (TAD) and transrepression (TRD) domains at the C-terminal, a region crucial for modulating HIF activity due to its multiple interactions with transcriptional coactivators and repressors [59].

HIF activity depends on the heterodimerization of its α and β subunits, and this interaction is regulated mainly by the availability and stabilization of HIF-α, which occurs under low oxygen levels. Under normoxic conditions, HIF-α is hydroxylated by the EGLN (EGL nine homolog) family, also referred to as prolyl hydroxylase domain-containing proteins (PHDs), and subsequently targeted for degradation. However, HIF activity can also be modulated by oxygen-independent mechanisms. For instance, oxidative stress can inactivate PHDs via ROS accumulation, and multiple noncoding RNAs regulate the stability and function of both EGLNs and HIFs. These and other oxygen-independent mechanisms will be discussed in more detail later. This complexity in HIF regulation reflects its broad functional scope and impact in various biological processes [60,61]. The oxygen-dependent degradation domain (ODD) in HIF-α mediates its degradation under normoxic conditions. This HIF-α destruction occurs when ODD is hydroxylated at specific proline residues (P402 and P564) by prolyl-4-hydroxylases (PHDs), a family of 2-oxoglutarate-dependent dioxygenases. These enzymes also require iron to hydroxylate HIF-α, facilitating its recognition and tagging by the von Hippel-Lindau protein (VHL), which recruits the E3 ubiquitin ligase complex (including Elongins B/C). This recognition and binding lead to the proteasomal degradation of HIF-α [58,59,62]. This degradation pathway is the most widely recognized and plays a dominant role in the HIF function. The activity of PHDs regulates HIF stability dynamically and is strongly influenced by the spatiotemporal oxygen gradient within the tissue [62].

Although PHDs are the main enzymes involved in oxygen-dependent regulation, HIF-α activity is also suppressed by other factors, such as Factor Inhibiting HIF (FIH). FIH irreversibly hydroxylates an asparagine residue (N803) near the C-terminal region of HIF-α. This change inhibits the interaction between HIF-α and transcriptional coactivators such as p300 (E1A Binding Protein p300) and CBP (CREB Binding Protein), which are essential for gene activation by HIF [62].

Several oxygen-independent factors, already introduced, contribute to the regulation of HIF activity. For instance, interaction with various kinases, such as ERK1/2 (extracellular signal-regulated kinases 1 and 2, also known as p44 and p42 MAPKs) and p38 MAPK (a mitogen-activated protein kinase activated by cellular stress and proinflammatory cytokines), stimulates HIF-α synthesis, thereby increasing its availability. Furthermore, in some cases, these kinases enhance HIF interaction with coactivators, such as p300. Another example is p53 (Tumor protein p53), a key tumor suppressor that reduces HIF-α degradation, thereby favoring its accumulation and activity [58,62].

Among oxygen-independent regulatory elements, oxidative stress and reactive oxygen species (ROS) can inhibit PHD activity through redox-mediated oxidation, leading to HIF-α stabilization. Although ROS is derived from oxygen metabolism, its effect on PHDs does not require direct oxygen sensing. Interestingly, sustained nitric oxide (NO) exposure inhibits mitochondrial respiration, reducing oxygen consumption and increasing its availability for PHD-mediated HIF-α degradation [60,61,62]. Additionally, transcription factors such as ATF-1 (Activating Transcription Factor 1) and CREB-1 (cAMP Response Element-Binding Protein 1) have been reported to bind cis-regulatory elements within HREs that may enhance HIF-driven gene activation.

Epigenetic conformation adds yet another layer of complexity to the regulation of HIF activity. Although many functional HREs are found in non-methylated CpG-rich regions, methylation of CpG sites within HREs can block HIF binding and specifically restrict the hypoxic response to accessible gene loci [61]. This epigenetic landscape can vary across different cell types. In some cases, selective pressure may prevent the methylation of key genes, such as erythropoietin (*EPO*), in specific cell types and developmental contexts [62]. Epigenetic conformation can also be modified by ROS, which oxidizes nucleotides in gene sequences such as *VEGF* and promotes HIF binding to their HREs [62]. Additionally, the activity of multiple microRNAs (miRNAs), such as miR-18a, miR-155, or miR-429, can inhibit hypoxia-inducible HIF-1α, while others like miR-145 suppress, and miR-558 enhance, the expression of hypoxia-inducible factor HIF-2α through direct interactions with EPAS1 (endothelial PAS domain-containing protein 1) mRNA, the transcript of the gene encoding HIF-2α [60].

Not only is HIF activity modulated by the complex regulation of HIF-α stability, but also by the distinct interactions and dynamics among its isoforms: HIF-1α, HIF-2α, and HIF-3α. These isoforms exhibit distinct spatiotemporal expression patterns and play non-redundant roles in cellular responses to hypoxia [60].

HIF-1α is considered the most ubiquitously expressed isoform and is primarily associated with acute hypoxia. It promotes rapid adaptive responses that allow the cell to survive immediate oxygen deprivation. These include metabolic reprogramming, stimulation of glycolysis, maintenance of pH balance, control of apoptosis, and the initiation of pro-angiogenic processes [60]. In contrast, HIF-2α is more prominently expressed in chronic hypoxia and supports long-term adaptation. It contributes to extracellular matrix remodeling by inducing matrix metalloproteinases, which are essential for angiogenesis and neovessel formation. Additionally, HIF-2α is involved in stem cell maintenance, cellular proliferation, antioxidant defense, and the stimulation of erythropoiesis via upregulation of *EPO* expression [60,63]. Figure 1 provides an overview of the oxygen-dependent regulation and functional divergence between HIF-1α and HIF-2α.

The effects and regulation of HIF isoforms are inherently complex. Numerous studies describe distinct transitions between HIF-1α and HIF-2α activity. In specific contexts, both isoforms act synergistically, as seen in the stimulation of angiogenesis through VEGF-A. However, in many others, their actions oppose each other and are strongly dependent on spatial and temporal conditions [60,64]. For example, in Human Umbilical Vein Endothelial Cells (HUVECs), 75% of HIF-1α target genes were activated within the first 2 h following oxygen reduction, whereas HIF-2α increased the expression of 80% of its target genes only after 8 h [64]. A similar pattern was observed in colon cancer cells, where only 196 genes changed expression under acute hypoxia, compared to 4149 genes affected under chronic hypoxia. Overall, this emphasizes the relevance of isoform timing in regulating hypoxia-driven cell fate [60].

Beyond this temporal and functional divergence, several molecular mechanisms differentially regulate each isoform. These include transcriptional coactivators, specific transcription factors, epigenetic conformation, isoform-specific mRNA stability under varying hypoxic conditions and oxidative stress environments, and interactions with multiple miRNAs. Some of these regulatory components are summarized in Table 2 [60,63,64]. All of these are crucial for fine-tuning hypoxic responses in various cellular environments.

HIF-3α joins this regulatory framework as the least explored, yet it plays an essential role in modulating the activity of the other HIF-α variants. It primarily acts as an inhibitor of genes activated by HIF-1α and HIF-2α. Multiple HIF-3α isoforms exist, including a truncated variant known as IPAS (Inhibitory PAS domain protein), which lacks the C-terminal region and the C-TAD (C-terminal transactivation domain) required to interact with transcriptional coactivators. This structure has been associated with a possible dominant-negative effect on the other isoforms, meaning it may competitively inhibit their function by interfering with dimerization or coactivator binding [60,63,65]. Interestingly, other HIF-3α isoforms have been reported to exert stimulatory and additive effects during early angiogenesis and in chronic hypoxia, in conjunction with HIF-2α. Overall, HIF-3α is predominantly characterized as an inhibitory modulator of HIF-1α and HIF-2α activity, particularly in endothelial and smooth muscle cells, likely through a negative feedback mechanism [60,63,65].

The complex regulation of HIF isoforms is not only essential for cellular adaptation but also plays a critical role in guiding stress responses and regenerative processes across multiple tissues.

### 3.2. Non-HIF-Dependent Mechanisms of Adaptation to Hypoxia

While HIF-dependent pathways play a central role in the cellular response to hypoxia, several non-HIF-dependent mechanisms also contribute significantly to cell survival, remodeling, and fate under low oxygen tension. These include alternative transcription factors, mitochondrial proteome remodeling, calcium handling through the Mitochondrial Calcium Uniporter (MCU) and its regulator Mitochondrial Calcium Uptake 1 (MICU1), the Mitochondrial Permeability Transition Pore (MPTP), adaptive metabolic shifts, and apoptotic pathways independent of HIF.

Among transcriptional regulators, Nrf2 acts independently of HIF under hypoxic and oxidative stress conditions. Upon activation, Nrf2 dissociates from its cytoplasmic inhibitor, Kelch-like ECH-associated protein 1 (KEAP1), translocates into the nucleus, and binds to antioxidant response elements (AREs), thereby promoting the expression of cytoprotective genes [66]. Similarly, the Integrated Stress Response (ISR), mediated by Activating Transcription Factor 4 (ATF4), can drive stress-response gene expression and metabolic adjustments that enhance homeostasis in hypoxic environments [67].

Hypoxia also triggers profound proteomic remodeling. In adipose-derived stem cells (pASCs) exposed to 1% O_2_ for 24 h, 361 proteins were differentially expressed, primarily associated with energy metabolism, mitochondrial function, and stress responses [68]. Comparable changes were observed in human intestinal organoids subjected to hypoxia and reoxygenation, favoring glycolytic metabolism and antioxidant defenses through ATF4 and ISR activation [67]. These findings highlight the ability of cells to reprogram their proteome through mechanisms not exclusively driven by HIF.

Mitochondria are central sensors and effectors of hypoxic adaptation. Calcium released from the endoplasmic reticulum enters mitochondria via the MCU. Excessive Ca^2+^ uptake can trigger the sustained opening of the MPTP, leading to mitochondrial depolarization, cytochrome c release, and apoptosis. Controlled Ca^2+^ entry and transient MPTP opening, instead, promote metabolic reprogramming, structural remodeling (fusion/fission), and cell survival [69]. Experimental evidence shows that MCU activity and MPTP resistance vary between rat strains with high (August) or low (Wistar) tolerance to acute hypoxia, contributing to differential adaptation [70,71]. Additionally, MICU1 acts as a critical gatekeeper for Ca^2+^ influx; its overexpression under hypobaric hypoxia restores mitochondrial homeostasis, reduces apoptosis and fibrosis, and improves cardiac function [72]. In the nervous system, MCU activity is essential for the neuroprotective effect of ischemic postconditioning in cerebral ischemia-reperfusion models, further highlighting its role in non-HIF-mediated adaptive responses [73].

Hypoxia can also modulate mitochondrial activity via post-translational modifications. Lactylation of mitochondrial proteins, such as Pyruvate Dehydrogenase E1 Alpha 1 (PDHA1) and Carnitine Palmitoyltransferase 2 (CPT2), reduces oxidative phosphorylation (OXPHOS), promoting a glycolytic shift as a rapid adaptation to limit oxygen consumption and ROS production [74].

Finally, severe hypoxia can trigger apoptosis through HIF-independent mechanisms. In human pluripotent stem cells, hypoxic stress activates caspases (CASP-9 and CASP-3) and disrupts the balance between pro-apoptotic BCL2-Associated X Protein (BAX) and anti-apoptotic Myeloid Cell Leukemia-1 (MCL-1), leading to cell death independently of HIF-1α and tumor suppressor protein p53 [75].

Collectively, these findings reveal the complexity of hypoxic regulation beyond HIF, underscoring the contribution of mitochondrial remodeling, calcium dynamics, alternative transcriptional pathways, and non-HIF-controlled apoptosis to cellular adaptation. Recognizing these complementary mechanisms is crucial for understanding interindividual differences in hypoxia tolerance and for identifying new therapeutic targets that enhance cell survival and tissue regeneration under low oxygen availability.

## 4. Hypoxia in Regenerative Processes

Hypoxia has a profound impact on processes such as angiogenesis, immune and inflammatory responses, as well as cellular proliferation and energy metabolism, all of which are critical for tissue repair. This influence can be either beneficial or detrimental, depending on the specific context and severity of the hypoxic exposure. This context raises the following question: What is the role of hypoxia in tissue regeneration processes? In many cases, tissue injury occurs in a local hypoxic context, while in others, systemic oxygen deprivation, whether acute, chronic, or intermittent, shapes the response to injury. Regenerative responses may vary depending on the tissue involved and the specific hypoxic context in which the injury occurs [15,62,76].

In recent years, this question regarding tissue regeneration under hypoxia has become increasingly central. These studies address it through various approaches, including bioengineering, in vitro, and in vivo models. The following section highlights key tissue contexts in which significant advances have been made. They reveal how hypoxia triggers both universal cellular pathways and context-dependent regenerative outcomes across tissues.

### 4.1. Cardiac Regeneration

The heart is vital in vertebrates, and its restoration after injury is critical. This regenerative process is very limited in mammals, including humans, in whom cardiovascular disease remains the leading global cause of morbidity and mortality. This problem is reflected in 18 million deaths annually and approximately 64 million people living with Heart failure and functional limitations [77]. In this context, improving regenerative processes has become a priority, motivating active research efforts. However, despite biomedical advances, myocardial regeneration in humans remains a significant challenge [78,79,80,81]. Recent studies have explored organisms with variable cardiac regenerative capacity, ranging from high to low, under hypoxic conditions or pharmacological induction of HIF stabilization. These studies have made it possible to explore the effect of hypoxia on regeneration, identifying differential elements and potential therapeutic approaches that regenerative medicine could apply in the future. Different animal models with variable regenerative capacities under hypoxia are analyzed below.

In models with high regenerative capacity, hypoxia acts as a key modulator of the process, as observed in zebrafish (*Danio rerio*) and African clawed frog (*Xenopus laevis*). In these organisms, the stabilization of HIF-1α stimulates the expression of transcription factors such as Isl1 (ISL LIM Homeobox 1) and Nkx2.5 (NK2 Homeobox 5), both of which are essential for cardiac tissue differentiation and repair. This more active response in these organisms may partially explain their regenerative capacity, in contrast to species with severe limitations, such as mice [82]. In zebrafish, cardiac regeneration is complete and driven by dedifferentiation and proliferation of preexisting cardiomyocytes [83]. This proliferative response increases under hypoxic conditions (13% O_2_) and after ventricular injury, with the percentage of proliferating cardiomyocytes rising from 3% to 6% compared to normoxic controls. This behavior is directly related to the activity of HIF-1α and VEGF [84].

Other animal models that are phylogenetically closer to humans have been explored. One of them is the mouse (*Mus musculus*). In this species, a remarkable phenomenon occurs: neonates at early stages exhibit a tremendous capacity to regenerate their heart, but this ability persists for only a short time after birth. This effect has been associated with natural hypoxia during intrauterine life [85,86]. The stabilization of HIF-1α in these early stages is crucial for inducing cardiomyocyte dedifferentiation and re-entry into the cell cycle, thereby promoting regeneration [85,86]. This regenerative effect is also linked to the stimulation that HIF-1α induces in anaerobic metabolism and cell proliferation. Thus, cell divisions are directly dependent on the repression of cell cycle suppressors, such as Cyclin-Dependent Kinase Inhibitor 1A (p21) and Cyclin-Dependent Kinase Inhibitor 1C (p57), as well as ATF4, during early cardiac development [87].

In contrast to the neonatal stages, proliferative activity persists in adult mice, although in a much more limited manner. This behavior was confirmed using a transgenic mouse model in which hypoxic cardiomyocytes and their proliferative activity could be tracked over time [88]. This study demonstrated that these cell populations are predominantly found in less perfused and hypoxic areas, where HIF-1α is stabilized, leading to increased activity of cyclins and CDK proteins, which are crucial for cell cycle progression, and resulting in reduced DNA damage. These populations supported an annual cardiomyocyte turnover of approximately 0.6% but became more active following induced myocardial ischemia [88]. Despite this increased cardiomyocyte activity, the turnover rate appears to be insufficient to repair injuries in adult mice.

In addition to these observations in tissue microenvironments, the effects of controlled systemic hypoxia have been studied in adult mice, showing a positive impact on cardiac recovery. This fact was observed by inducing progressive systemic hypoxia, reaching 7% O_2_. This condition triggered significant cellular changes, including reduced mitochondrial activity and oxidative stress, which preserved DNA integrity. These effects were associated with increased cardiomyocyte proliferation, supporting recovery after ischemia [89].

Preconditioning and postconditioning, which induce the stabilization of HIF-1α and HIF-2α, applied in short intervals, have also shown beneficial effects after coronary ischemia in adult mice. These conditioning periods, especially those applied before injury, promoted the activity of HIF-1α and HIF-2α, resulting in improved myocardial recovery and a smaller infarct area compared to the control group [90]. These beneficial effects are consistent with the actions of HIF-1α in cardiomyocytes, which include the reactivation of angiogenesis, a shift toward anaerobic metabolism, and a reduction in apoptosis [91]. In addition to these mechanisms, HIF-1α also promotes an increase in Interleukin-22 (IL-22), a cytokine that supports cardiomyocyte proliferation and survival. These effects are mediated by the B-cell-specific Moloney murine leukemia virus integration site 1 (Bmi1) transcription factor [92].

The long-term effects of sustained activation of these factors on cardiac homeostasis have also been investigated. The chronic impact of HIF-2α has been explored in mice. For example, in a murine model exposed to chronic hypoxia (10% O_2_), the conditional deletion of HIF-2α in vascular cells (Wt1 lineage) resulted in cardiomegaly, ventricular hypertrophy, dilation, and systolic dysfunction, indicating that vascular signaling mediated by HIF-2α contributes to preserving cardiac homeostasis under chronic hypoxia [93]. Moreover, a sustained overexpression of HIF-2α promotes cardiomyocyte proliferation, minimizes oxidative stress, and prevents DNA damage. These effects result in reduced fibrosis and improved systolic function following myocardial infarction in mice [94].

However, the effects of hypoxia must be carefully regulated. Persistent activity of HIF-1α and HIF-2α can be harmful, leading to ventricular dilation and decreased ejection fraction. This situation has been described following prolonged stabilization of HIF-1α in transgenic mice. In these animals, inhibition of Sarco/Endoplasmic Reticulum Ca^2+^ ATPase (SERCA), a key enzyme for calcium homeostasis in cardiomyocytes, was observed. Reduced SERCA activity ultimately contributes to ventricular dysfunction [95].

The previously described findings, along with other recent results, reveal a significant level of contradiction. As an additional example, contrasting outcomes have been reported in studies using transgenic mouse models. On the one hand, enhanced recovery and vascularization were observed after myocardial injury in mice that constitutively expressed stable HIF-1α from early developmental stages [96]. On the other hand, another study reported metabolic alterations and disrupted calcium homeostasis in cardiomyocytes, leading to late-onset cardiomyopathies in mice with HIF-1α overexpression [97]. These maladaptive effects are particularly evident in chronic exposure models that simulate extreme hypoxia, often equivalent to altitudes of 8000 m. Under these conditions, cardiac overload, pulmonary hypertension, and ventricular hypertrophy are commonly observed [98]. These findings highlight the dual nature of hypoxia effects and the importance of precise regulation to achieve regenerative rather than maladaptive outcomes. This variability underscores the need to standardize experimental conditions to enable meaningful comparisons between studies [99].

Although research in human patients faces greater limitations, a recent study showed that blood oxygen saturation levels (SaO_2_), which reflect the availability of this gas in tissues, may influence cardiac regeneration. By comparing different SaO_2_ levels in patients, it was reported that values ranging between 75% and 85% optimally activated Yes-associated protein 1 (YAP1). This transcription factor promotes cardiomyocyte cell cycle reactivation, thereby providing insight into the increase in cell proliferation within a still minimal regenerative context [100].

Therapeutic strategies based on hypoxia-preconditioned stem cells, extracellular vesicles, and biomaterials have also been explored to enhance cardiac repair, as detailed in Section 5.1. At the cellular level, other complex intracellular mechanisms also regulate the hypoxic response during cardiac regeneration.

In vitro studies using neonatal rat ventricular myocytes (NRVMs) and adult feline cardiomyocytes have shown that Uncoupling Protein 2 (UCP2), overexpressed under moderate hypoxia (5% O_2_), promotes mitochondrial uncoupling and a metabolic shift toward glycolysis. It also modulates acetyl-CoA levels and histone acetylation to favor cardiomyocyte proliferation. These actions, not directly linked to HIF activity, represent alternative adaptations induced by hypoxia during cardiac regeneration. Similarly, in an in vivo murine model exposed to 7% O_2_ for four weeks, increased UCP2 expression was essential for maintaining ventricular function and limiting fibrosis. In contrast, its absence was associated with greater myocardial damage [101]. Furthermore, HIF-1α stabilization may depend on complex regulatory mechanisms, such as the Forkhead box protein P1 (Foxp1)–Ubiquitin-specific peptidase 20 (Usp20)–HIF-1α axis. In both in vivo and in vitro murine models, the suppression of Foxp1 promoted HIF pathway activation and enhanced myocardial regeneration [102].

Figure 2 summarizes the main differences in cardiac regenerative capacity between some of the species studied and humans described previously, whereas Figure 3 shows the hypoxia-induced cardiac responses, highlighting regeneration in experimental models and cardioprotection in humans.

In conclusion, evidence from various organisms and in vivo models, as well as pharmacological and biomedical approaches conducted in vitro and in vivo, demonstrates that hypoxia and HIF stabilization, when applied under controlled conditions of duration and severity, can promote regenerative responses in cardiac tissue. These beneficial effects are attributed mainly to the induction of angiogenesis, cardiomyocyte proliferation, and enhanced cell viability in response to stress or injury. As outlined in Section 5.1, complementary strategies utilizing hypoxia-preconditioned cells and their derivatives may further potentiate these regenerative processes, highlighting the need for precise regulation to prevent maladaptive outcomes.

### 4.2. Muscle Regeneration

Skeletal muscle regeneration is a complex process highly dependent on the activity of resident stem cells and their ability to respond to environmental cues such as oxygen availability. There is consensus regarding the positive effect of HIF on the self-renewal of satellite cells, a population of muscle-resident stem cells involved in tissue repair and regeneration. However, satellite cell self-renewal appears to compete with the differentiation and maturation of muscle fibers, which are essential for effective regeneration. Experimental and in vitro models, including murine C2C12 myoblasts and human or porcine mesenchymal cells, suggest that hypoxia enhances migration and proliferation while variably affecting differentiation, depending on signaling pathways such as the NOTCH and non-canonical Wnt pathways [103,104,105,106]. Detailed findings on preconditioning protocols are expanded in Section 5.1.

Additionally, there is evidence that HIF-2α contributes to maintaining undifferentiated states and promoting stem cell self-renewal, in close association with the expression of the *Spry1* gene (Sprouty homolog 1) [107]. HIF-1α has also been implicated, particularly in mice, where it has been linked to reduced activity of the PI3K/Akt pathway, an essential signaling axis for the activation of myogenic factors [108].

Hypoxic conditions play a key role in determining the outcomes of muscle regeneration. For example, intermittent hypobaric hypoxia (4500 m; ~12.5% O_2_ equivalent) has shown beneficial effects on muscle regeneration in rats, which also exhibited reduced fibrosis. These data were correlated with increased Akt/mTOR activity and the expression of AMP-activated protein kinase (AMPK). This cellular energy sensor regulates metabolism, mitochondrial function, and autophagy in response to energy stress [109].

These results contrast with findings from rats exposed to simulated hypobaric hypoxia (5500 m; ~11.2% O_2_ equivalent), where regeneration was initially impaired due to reduced mTOR activity. However, after 28 days, recovery levels were equivalent to those observed in control groups [110].

Similarly, the positive effects of acute hypoxia, observed during 168 h following muscle injury in mice, were associated with the stimulatory role of HIF-1α in angiogenesis (via VEGF), macrophage migration, and metabolic adaptation [111]. This isoform also promotes immune processes by inducing the expression of inducible nitric oxide synthase (iNOS). This enzyme produces nitric oxide in response to inflammatory or hypoxic stimuli, contributing to immune responses and tissue regeneration [112].

Among the factors involved in muscle regeneration, VEGF plays a vital role. VEGF upregulation has been implicated not only in the neovascularization of regenerating muscle tissue but also in the activation of a higher number of satellite cells at the injury site, reduced muscle fiber necrosis, and increased muscle strength following damage [113,114]. This fact suggests that HIF-induced VEGF expression may also contribute to these regenerative processes.

While transient hypoxia, when applied in the context of acute muscle injury, has been shown to promote regeneration, primarily through the activity of HIF-1α. Chronic and intense hypoxia, as observed in muscle diseases, is linked to increased fibrosis. This process is dependent on the connective tissue growth factor (CCN2) (also known as CTGF) and transforming growth factor beta (TGF-β). These adverse outcomes have also been associated with an increased activity of the pro-inflammatory molecule angiotensin II [115]. For example, under chronic hypobaric hypoxia (PpO_2_ 15%; ~5500–5800 m), a reduction in satellite cell numbers and impaired activation were observed, compromising long-term muscle integrity. This effect was associated with HIF-2α stabilization and more vigorous activity of the Renin-Angiotensin-Aldosterone system (RAAS), contributing to vascular dysfunction and inflammation [116].

Together, these findings support a scenario in which hypoxia promotes satellite cell self-renewal while hindering terminal fiber differentiation. It also induces metabolic shifts toward glycolysis and promotes early vascularization. However, the outcome of hypoxia appears to be highly dependent on its severity and chronicity.

### 4.3. Bone Regeneration

Bone regeneration is a highly orchestrated process that depends on the balance between bone formation, resorption, and vascularization, processes tightly regulated by local oxygen availability and HIF signaling pathways. Bone remodeling involves the coordinated activity of osteoblasts, which build new bone; osteoclasts, which resorb it; and endothelial cells, which provide vascular support essential for tissue repair. However, existing data show both positive and negative effects, and in some cases, even contradictory findings regarding the impact of hypoxia on bone integrity. These effects have been explored through bioengineering with scaffolds, as well as in vitro, ex vivo, and in vivo models. The variety of experimental models and conditions plays a key role in evaluating these contrasting results.

In vitro studies using stem cell cultures under pharmacological stabilization of HIF with Dimethyloxalylglycine (DMOG) have shown increased expression of osteogenic markers, including collagen type I (COL1, a major structural protein of the bone matrix), Runt-related transcription factor 2 (RUNX2, a transcription factor essential for osteoblast differentiation), osteocalcin (OCN, a non-collagenous protein secreted by osteoblasts during bone mineralization), and alkaline phosphatase (ALP, an enzyme associated with early stages of bone formation) [117]. Similar effects have been reported when DMOG was combined with hydrogels, promoting endochondral differentiation [118]. Detailed findings on hypoxia-based preconditioning approaches, including DMOG-treated stem cells and biomaterial-assisted delivery systems, are expanded in Section 5.1. Beyond these preconditioning applications, other studies have investigated the broader effects of hypoxia on bone remodeling under pharmacological induction, particularly through HIF stabilization with DMOG, as summarized below.

Several studies have shown the beneficial effects of hypoxia. Some of them are described under pharmacological induction and stabilization of HIF using DMOG, a prolyl hydroxylase inhibitor that stabilizes HIF. For instance, in vitro stem cell cultures treated with DMOG exhibited increased expression of osteogenic markers, including collagen type I (COL1), Runt-related transcription factor 2 (RUNX2), osteocalcin (OCN), and alkaline phosphatase (ALP), all of which are associated with osteoblast differentiation and mineralization [117]. In a similar context, DMOG has also been incorporated into hydrogels, where it enhances endochondral differentiation—a process that first forms cartilage and then bone, making it potentially useful for bone repair. Although this process may be slow, it has the potential to be useful for bone restoration [118].

These effects have been associated with increased angiogenic activity via vascular endothelial growth factor (VEGF) and the stimulation of mesenchymal cells, derived from various species and tissues, to integrate into osteogenic processes in vitro [119].

Building on these findings, further studies have investigated the use of DMOG in combination with various cell types in preclinical models. These included adipose-derived stem cells from rats [120] and bone marrow-derived mesenchymal stem cells (BMSCs) from young rats that were pre-treated with DMOG and cultured under hypoxic conditions, to enhance osteogenic and angiogenic responses [121]. These approaches demonstrated improved vascularization as a result of VEGF signaling activation, osteogenic differentiation, and increased expression of cell markers, including RUNX2 and OCN. They also improved and accelerated bone healing in calvaria and mandibular defect models in rats, respectively [120,121]. Similar osteogenic stimulation, evidenced by increased ALP activity, was observed in rat periodontal defects treated with DMOG-based grafts [122].

Beyond structural effects on osteogenic differentiation, hypoxia may also exert modulatory roles in the bone immune environment. It promotes the polarization of macrophages toward the M2 phenotype, which supports tissue renewal and repair while exerting anti-inflammatory functions [117]. All these positive effects of hypoxia, induced pharmacologically through DMOG, on bone formation, healing, and maintenance are widely discussed in a recent systematic review [117].

Vascularization plays a central role in bone formation, with VEGF acting as a key stimulator of this process. The VEGF–HIF axis has also been explored in bone regeneration. This angiogenic effect of VEGF was demonstrated in vitro using Salidroside (SAL), a pharmacological stabilizer of HIF. In this study, SAL-treated cultures exhibited enhanced function and promoted endothelial sprouting, thereby contributing to the osteogenic process [123].

Despite the evidence of beneficial roles of hypoxia in osteogenesis and vascularization, other studies have reported adverse effects under certain conditions [124,125]. For example, primary rat osteoblasts cultured under decreasing oxygen concentrations, from standard 20% O_2_ (normoxia in vitro, but hyperoxia relative to physiological bone levels) to less than 1% O_2_ (moderate to severe hypoxia), showed reduced mineralization, differentiation, proliferation, and ossification. They also showed disrupted collagen organization and impaired bone nodule formation. All of this was accompanied by a decrease in the expression of markers such as RUNX2, ALP, and OCN [126].

Consistent with these findings, a more deleterious effect of glucocorticoids on bone and osteoblasts was reported under hypoxic conditions. In this case, increased apoptosis and inhibition of osteogenesis were associated with enhanced activity of the phosphoinositide 3-kinase (PI3K)/protein kinase B (Akt) signaling pathway, a key regulator of cell survival, growth, and metabolism, stimulated by HIF [127].

A subsequent study explored both in vitro and in vivo models. In vitro, hypoxic culture conditions altered osteoclast behavior. In vivo, the osteogenic evaluation of mice with knockdown of prolyl hydroxylases (PHDs), resulting in the sustained stabilization of HIF, demonstrated a marked increase in osteoclast presence and enhanced trabecular bone resorption [128].

Particularly, HIF-2α activity has been linked to these effects. This activity was demonstrated by evaluating mice with either deficiency or overexpression of HIF-2α. The activity of this isoform correlated with bone resorption and reduced bone mass. All of this was accompanied by reduced expression of RUNX2 and OCN, and increased expression of receptor activator of nuclear factor kappa-B ligand (RANKL), which promotes osteoclast maturation [129]. Beyond these opposing effects, specific experimental contexts present further contradictions.

This controversy becomes evident under specific conditions, such as the effects of hypobaric hypoxia on bone. A recent review proposed an adverse effect of chronic exposure to altitudes above 2500 m [124]. Sush effect was supported by a potential increase in erythropoietin (EPO), which stimulates erythropoiesis. The resulting polycythemia and increased blood viscosity could impair bone perfusion. Likewise, oxidative stress, mitochondrial dysfunction, and a tendency toward acidosis have been identified as contributors to bone resorption and increased osteoclastic activity [124].

However, these findings have been challenged by a recent study using a model of controlled hypobaric hypoxia (50 kPa; 10–11% O_2_), applied in both chronic and intermittent regimens. This study evaluated the recovery of femoral bone defects in rats. In this work, accelerated recovery was reported under hypoxia, with bigger callus formation, remodeling, and improved biomechanical properties of the newly formed bone. All of this occurred in the context of increased angiogenesis and expression of VEGF, osteogenic markers such as RUNX2, SP7 (Sp7 transcription factor, also known as Osterix), and collagen type I (COL1), and proteins associated with mesenchymal cell migration and repair, including stromal cell-derived factor 1 (SDF-1) and C-X-C chemokine receptor type 4 (CXCR4) [130].

Other controversial findings concern the effect of EPO. Some publications support the idea that EPO promotes bone resorption and, as previously mentioned, reduces vascular perfusion to bone [124]. This latter effect is mainly attributed to polycythemia, with or without EPO overactivity [131]. Another study associated the adverse effects of EPO, both in vitro and in vivo using mice, with the specific stimulation of osteoclasts. These cells enhanced resorption through the Janus kinase 2 (JAK2) pathway, a protein involved in cytokine signaling and hematopoietic regulation, as well as the PI3K pathway, depending on the EPO dose involved [132].

These data are challenged by an in vitro study using human bone marrow stromal cells (hBMSCs), which instead showed a greater osteogenic response. This behavior was evidenced by increased ALP activity and activation of the protein kinase B (Akt)/mechanistic target of rapamycin (mTOR) pathway, a signaling pathway that promotes cell growth, proliferation, and survival [133].

Similarly, two other studies reported in vivo experiments in mice with femoral osteotomy, treated with EPO and evaluated over a 2–10 week period following fracture. In both cases, a positive effect of EPO was observed, with faster recovery, increased callus formation and vascularization, and higher VEGF expression, as well as ossification and biomechanical strength [134,135].

Taken together, these findings suggest that hypoxia has contrasting effects depending on the specific conditions and experimental context. Therefore, continued investigation in this area is essential, using diverse in vitro and in vivo models, as this phenomenon holds significant biomedical potential. Interestingly, similar dual outcomes of hypoxia have been reported in muscle regeneration, which are addressed in the following sub-section.

### 4.4. Hypoxia and Vascular Responses During Regeneration

Angiogenesis is a critical early event in tissue regeneration, as sufficient perfusion is essential for repair. Hypoxia and neovascularization are tightly interconnected. Across different tissues, hypoxic conditions activate VEGF signaling and other pro-angiogenic pathways, which restore blood supply to damaged areas and contribute to the regenerative process.

In skeletal muscle, hypoxia strongly stimulates angiogenesis, supported by the involvement of M2 macrophages in vascular remodeling and contractility [136]. The activity of M2 macrophages illustrates how the hypoxic microenvironment can orchestrate vascular changes that favor tissue repair, beyond its direct effects on myogenic cells.

In liver repair, pharmacologically induced hypoxia has been reported to increase VEGF-mediated neovascularization, improving perfusion and preserving the fenestrated morphology of hepatic sinusoidal endothelial cells [137,138]. These vascular effects appear to result not only from hypoxic signaling itself but also from increased activity of the NOTCH pathway in regenerative settings [139]. This finding suggests that vascular responses to hypoxia are not limited to skeletal muscle but also extend to specialized tissues, such as the liver, where improved blood flow is crucial for regeneration.

Similar vascular mechanisms have also been observed in neural regeneration models, underlining the conserved role of hypoxia-driven angiogenesis across tissues. Extracellular vesicles (EVs) derived from hypoxic microglia contribute to vascular remodeling, improve nerve perfusion, and support post-ischemic structural repair [140,141]. Moreover, VEGF supplementation was necessary to sustain vascular support for nerve repair when HIF activity was experimentally reduced under normoxia [142].

With the global view, we can conclude that hypoxia universally triggers angiogenic responses that support regeneration across tissues, primarily mediated by VEGF signaling, macrophage polarization, endothelial activation, and vascular remodeling.

### 4.5. Hypoxia in Appendage Regeneration

Beyond its effects on individual tissues, hypoxia also plays a critical role in the regeneration of complex structures that integrate multiple tissue types, such as osteomuscular, neural, and vascular components. One example of a complex structure is the regeneration of appendages such as tails or limbs. This process has been studied in tadpoles (*Xenopus laevis),* where HIF-1α plays a crucial role by interacting with the expression and activity of the Wnt signaling pathway, which is closely related to early developmental processes, including the maintenance of posterior polarity and positional identity in developing limbs and appendages [143]. The reportin (*Xenopus laevis*) is consistent with findings in house geckos (*Hemidactylus platyurus*) after tail amputation. In these animals, an early stabilization pattern of HIF-1α is observed, promoting inflammation, blastema formation, and cellular activation and proliferation, followed by a more gradual and sustained expression of HIF-2α, associated with angiogenesis and cellular differentiation [144].

Comparative studies across species further underscore the role of hypoxia-driven mechanisms in contributing to regenerative capacity. In recent data comparing Xenopus laevis at pre- and post-metamorphic stages and axolotls (Ambystoma mexicanum), it has been shown that after hindlimb amputation, axolotls, followed by early-stage tadpoles and adult *Xenopus laevis*, exhibit increasing levels of HIF-1α stabilization in that order. This stabilization is correlated with enhanced glycolytic activity, associated with the expression of genes such as *pfkfb3* and *ldha*, as well as vegfa-mediated angiogenesis and increased cell proliferation. All these processes are implicated in the greater regenerative capacity, which is directly influenced by hypoxia [145] preprint.

In addition to transcriptional regulation via HIFs, reactive oxygen species (ROS) also appear to play a regulatory role in regeneration under hypoxic conditions. A local increase in ROS levels has been described in the tail amputation site of tadpoles (*Xenopus laevis*), especially within the first 24 h post-amputation. ROS promote the expression of genes such as *notch1*, *msx1*, *fgf10*, and *wnt5a*, which are involved in key regenerative processes, including progenitor cell reactivation, proliferation, polarity, and tissue remodeling [146]. A similar regenerative role of ROS is observed in axolotls (*Ambystoma mexicanum*), where limb amputation leads to early and localized ROS production, promoting the expression of genes such as *msx2*, *cdx4*, and *fgf20*, which are crucial for blastema formation. This gene expression is accompanied by the activation of pathways such as MAPK/ERK, p38, and AKT, associated with cell proliferation, differentiation, and tissue remodeling [147]. Although these two studies do not directly mention or evaluate hypoxia or HIF-α stabilization, a connection between ROS and HIFs can be inferred. This association is explained by the fact that tissue injury leads to reduced perfusion and thus local hypoxia, which promotes mitochondrial ROS production, also ROS that, as previously discussed in earlier sections, can modulate and stabilize HIF-α even in an oxygen-independent manner.

Altogether, these findings highlight that hypoxia, either directly or via ROS signaling, plays a central role in coordinating multiple regenerative mechanisms in complex structures, contributing to their capacity for restoration across species.

Taking together all regenerative processes described previously, Figure 4 provides an overview of the context-specific effects of different hypoxia types on tissue regeneration, highlighting both beneficial and maladaptive responses across multiple organs and structures.

## 5. Hypoxia in Regenerative Medicine

### 5.1. Stem Cells and Bioengineering Applications

#### 5.1.1. Heart

Hypoxia has been explored as a preconditioning tool to enhance the therapeutic efficacy of stem cells and their derivatives for cardiac regeneration. The following paragraphs summarize current findings on extracellular vesicles (EVs), microRNAs, circular RNAs, and biomaterial-assisted delivery systems obtained from hypoxia-preconditioned cells.

Among these approaches, recent studies have focused on intercellular communication through EVs released by human induced pluripotent stem cells (hiPSCs) cultured under 5% O_2_ (EV-H5). These EV-H5 are enriched with antioxidant proteins regulated by the KEAP1/NRF2 (Kelch-like ECH-associated protein 1/Nuclear factor erythroid 2-related factor) pathway, including PRDX1 (peroxiredoxin-1), PRDX6 (peroxiredoxin-6), and GSTP1 (glutathione S-transferase pi 1). When transferred to cultured human induced cardiomyocytes subjected to ischemia and reoxygenation, they activate NRF2 and increase the transcription of the antioxidant genes HMOX1 (heme oxygenase 1), SOD2 (superoxide dismutase 2), and CAT (catalase), thereby preserving cell integrity, contractility, and survival [148].

The latest reports, to date, suggest that short-term hypoxic exposure of hiPSCs before EV isolates can enhance their cardioprotective properties in vivo and in vitro in mouse models. Another study found that these EV-H5 also contain miR-302b-3p. This microRNA can silence the TGF-β/SMAD2 signaling pathway by reducing the expression of SMAD2 and TGFBR2. Since this signaling pathway promotes fibrosis and cardiac remodeling, the administration of miR-302b-3p in vitro and in vivo mouse models reduced fibroblast activity and fibrotic activity [149]. In a murine model of fibrosis induced by angiotensin II, exposure to EV-H5 reduces inflammation and collagen deposition, confirming their antifibrotic effect in vivo [149].

Until this point, preconditioning cells with moderate hypoxia before EV harvesting has been shown to modulate both antioxidant defenses and fibrotic responses in damaged myocardium. Likewise, various miRNAs whose expression is stimulated under hypoxia (e.g., miR-199, miR-210, miR-424) are carried in extracellular vesicles and act as key mediators of communication between cardiomyocytes and endothelial cells, promoting angiogenesis and myocardial proliferation. Conversely, others (such as the miR-15/miR-24 family) inhibit these processes, highlighting the complex control exerted by microRNA signaling during cardiac regeneration [150]. Circular RNAs, such as circWhsc1, induced by hypoxia and exported in EVs by murine cardiac endothelial cells, activate the Tripartite Motif-containing protein 59, Signal Transducer and Activator of Transcription 3–Cyclin B2 (TRIM59–STAT3–Cyclin B2) pathway, enhancing G_2_/M progression and mitosis of cardiomyocytes in vitro. In vivo, endothelial overexpression or EV-based delivery of circWhsc1 improves ejection fraction, reduces fibrosis, and enhances angiogenesis in adult C57BL/6J mice after infarction, whereas circWhsc1 inhibition delays regeneration in both adult and neonatal mice. Altogether, circWhsc1 synchronizes myocardial proliferation and neovascularization, promoting cardiac repair after injury [150]. Those evidences demonstrate that hypoxia-induced non-coding RNAs packaged in EVs can act as potent regulators of myocardial regeneration, offering a cell-free therapeutic potential application.

Beyond the previously described cellular strategies and intercellular communication mechanisms, pharmacological and tissue engineering approaches are also being developed to promote cardiac regeneration. One example involves the use of hydrogels loaded with HIF-1α stabilizers. These materials enhance angiogenesis and tissue repair in rats (Rattus norvegicus) subjected to ischemic injury [151]. Another strategy relies on the activation of HIF-1α, either through its overexpression in iPSCs differentiated into cardiomyocytes or through ischemic preconditioning in murine models. These activated cells release higher levels of pro-angiogenic factors, including VEGF, Angiopoietin-1 (Ang-1), Fibroblast Growth Factor 1 (FGF-1), and Platelet-Derived Growth Factor Receptor Alpha (PDGFRA). These molecules contribute to improved myocardial perfusion and tissue recovery after infarction in mice [90,152].

These findings show that combining hypoxia-based preconditioning of cells with biomaterial scaffolds or HIF-1α stabilizers can enhance their regenerative potential and support myocardial repair.

Collectively, the evidence summarized in this subsection shows that hypoxia preconditioning of stem cells and their derivatives (EVs, non-coding RNAs, and bioengineered constructs) improves their therapeutic impact on the injured heart in in vivo and in vitro models. However, further studies should clarify how different exposure times (short, interim, or long-term hypoxia) influence the molecular cargo and efficacy of these therapies, enabling the design of precise and standardized clinical applications.

#### 5.1.2. Muscle

Similar preconditioning strategies have been investigated for muscle-derived stem cells to improve their therapeutic potential after transplantation. In murine C2C12 myoblasts in vitro, hypoxia enhances migration and proliferation while preserving a progenitor-like state marked by Pax7 (Paired Box 7) expression, which impairs terminal myogenic differentiation [103]. Similar findings have been reported in human and porcine mesenchymal cells [104], as well as in vitro myoblast cultures and in in vivo assays in mice [104,105]. This poor differentiation may be explained by the increased activity of the NOTCH signaling pathway, a key regulator of stem cell fate and self-renewal, and by reduced stimulation of the non-canonical Wnt pathway, which modulates cell polarity, migration, and lineage commitment under hypoxic conditions [106,153].

Interestingly, these effects appear to depend on the severity of hypoxia. Exposure to very low oxygen levels (<1% O_2_) during preconditioning has been associated with improved differentiation and enhanced expression of myogenic markers, as highlighted in a recent systematic review [106]. These findings suggest that both the duration and level of hypoxia are critical determinants of the functional outcome of preconditioned muscle-derived cells.

#### 5.1.3. Bone

In vitro cultures of stem cells treated with DMOG, a pharmacological hypoxia inducer, have demonstrated increased expression of osteogenic markers, including collagen type I (COL1), Runt-related transcription factor RUNX2, OCN, and alkaline phosphatase ALP [117]. These molecules are directly involved in osteoblast differentiation, mineralization, and matrix formation.

Similar findings were observed when DMOG was incorporated into hydrogels, which promoted endochondral differentiation —a process that first generates cartilage and subsequently bone, suggesting potential applications of hypoxia in bone repair [118]. These approaches show that pharmacological stabilization of HIF can prime stem cells toward an osteogenic lineage even before transplantation, potentially improving their integration and function in vivo.

Building on these reports, other preclinical studies have combined DMOG treatment with various stem cell types, such as adipose-derived stem cells and bone marrow-derived mesenchymal stem cells (BMSCs), cultured under hypoxic conditions before implantation. This strategy enhanced VEGF-mediated angiogenesis, osteogenic differentiation, and the expression of RUNX2, OCN, and ALP, accelerating bone healing in calvarial, mandibular, and periodontal defect models in rats [120,121,122]. Together, these results suggest that the timing and method of hypoxia-based preconditioning, whether through direct low-oxygen exposure or pharmacological induction treatment, can significantly influence the regenerative performance of bone-targeted stem cell therapies.

#### 5.1.4. Nervous System

In neural tissue, mesenchymal stem cells (MSCs) preconditioned under hypoxia exhibit improved viability and expansion in vitro, promoting neuronal differentiation and supporting nerve repair and myelination when transplanted into rats following sciatic nerve injury [154].

Extracellular vesicles (EVs), which have demonstrated regenerative potential in multiple tissues, also play a significant role in neural repair. EVs derived from hypoxic microglia stimulated the polarization of M2-type microglia and activated the TGF-β1/Smad2/3 signaling pathway, a regulatory cascade involved in anti-inflammatory signaling and neuronal regeneration. These effects promoted neuronal survival, vascular remodeling, and post-ischemic regeneration [140]. Comparable benefits have been reported for hypoxia-derived EVs in both in vitro and in vivo models, including improved nerve perfusion and structural repair [104]. VEGF supplementation was necessary to sustain vascular support for nerve repair when HIF activity was experimentally reduced under normoxia [142].

Beyond cell-based and EV-mediated therapies, intermittent hypoxia has been investigated as a preconditioning approach to enhance neural resilience. Exposure to intermittent hypoxia increased the expression of *neuroglobin* (*Ngb*), cellular oncogene *Fos* (*c-Fos*), and *growth-associated protein 43* (*GAP-43*), a protein essential for axonal growth, remodeling, and regeneration. These molecular changes were associated with decreased apoptosis, maintenance of mitochondrial integrity, increased neuronal proliferation, and improved functional recovery following ischemic injury. These findings align with growing evidence on the neuroprotective role of hypoxic preconditioning and intermittent hypoxia in aging brain and stroke recovery [155,156].

Taken together, these studies indicate that hypoxia-based preconditioning, whether applied directly to stem cells or via extracellular vesicles, as well as systemic intermittent exposure, enhances neural survival, differentiation, and repair potential, supporting its future use in regenerative medicine for nervous system injuries.

### 5.2. Clinical Applications

While in vitro and in vivo approaches in preclinical animal models are well-developed, clinical data and human studies assessing the effects of hypoxia on tissue regeneration and protection remain scarce. In the clinical field, acute intermittent hypoxia (AIH)—characterized by short hypoxic cycles interspersed with normoxic periods—has been the most frequently evaluated approach. Some of the evidence in this regard is summarized below.

AIH, applied as five cycles per day of 6 min at an FiO_2_ of 12.6%, interspersed with 4-min normoxic intervals over a total of 14 days, was evaluated in healthy young men as a preconditioning strategy before a sustained hypoxic challenge (FiO_2_ 12.6% for 8 h). Individuals pre-exposed to AIH reported fewer symptoms of acute mountain sickness, improved cognitive performance, and better biomarker profiles compared to the control group [157]. This work represents an interesting approach in physiologically healthy individuals and points out the benefit of preconditioning, closely mirroring effects observed in vivo animal models.

When AIH is evaluated in regenerative processes, clinical studies have primarily focused on neuroprotection and neuroregeneration. Several studies report beneficial effects of AIH. For example, in patients with chronic tetraplegia, who were exposed to different treatments including AIH (FiO_2_ 0.09 for one minute, followed by normoxia for one minute, repeated for 15 cycles), sustained hypoxia (FiO_2_ 0.09 for 15 continuous minutes), showed a 91% increase in maximum voluntary strength and improved spinal motoneuron excitability compared to their own baseline and control treatment with normoxia (FiO_2_ 0.21 for 15 min) [158]. A similar approach was used to evaluate 50 patients with chronic stroke. The hypoxic protocol consisted of 15 cycles of FiO_2_ 0.09 for 1 min, followed by FiO_2_ 0.21 for 1 min, with subsequent assessments of gait performance. This intervention did not show global differences compared to conventional treatment without AIH. Nevertheless, in 42% of patients, performance in the tests improved significantly compared to their baseline functional status, potentially related to enhanced neural plasticity. Although no clear predictors of response were identified, interindividual variability should be kept in mind [159].

In another case series study, which included ten patients with chronic hemiparesis after stroke, the researchers evaluated the effect of four AIH sessions. Thus, the oxygen concentration in the airway provided was progressively reduced from 21% to 9% across sessions. Afterwards, patients treated showed improvements in grip strength and elbow flexion in the affected limb. Those results were linked to increased synaptic plasticity, enhanced motor neural drive, and increased recruitment of motor neurons [160]. Similarly, beneficial effects were observed in patients with incomplete cervical spinal cord injury. A single session of AIH of reduced concentration oxygen airway provided in 15 cycles of 60 s of 10.5% FiO_2_, with 60 s of 21% FiO_2_ within, enhanced corticospinal excitability and increased grip strength by up to 23% compared to conventional normoxic treatment [161].

Benefits in reducing blood pressure and anxiety levels in patients with acute stroke treated with AIH have also been reported. Under a hypercapnic-hypoxic environment (10% CO_2_ and 10% O_2_ for 3 min daily over 14 days)

Those patients improved significantly compared to those receiving conventional care. These outcomes were hypothesized to result from enhanced cerebral perfusion [162].

Although AIH appears promising in neuroregeneration, the significant variability in experimental protocols, together with interindividual response variability associated with multiple factors such as genotypic background, sex, lesion type, and severity, can lead to heterogeneous results, reduced reproducibility, and low viability of clinical translation. These findings underscore the importance of standardized protocols and the need for phase II and III clinical trials with larger and more representative patient populations. This issue becomes particularly critical given the dual and sometimes opposing effects of hypoxia [163].

Beyond the nervous system, AIH has also been explored in cardiovascular contexts. Various AIH protocols have been employed, including FiO_2_ levels of 0.10–0.14 for 1– to 5-min cycles per day over 2– to 4–week periods. Patients receiving such treatments either before or after a cardiovascular injury have shown hemodynamic and exercise tolerance improvements. These effects have been attributed to autonomic sympathetic stabilization and have been recently reviewed [164].

While clinical applications of AIH typically regulate FiO_2_ in controlled cycles, an alternative approach—remote ischemic preconditioning (RIPC)—induces short cycles of limb ischemia followed by reperfusion. This strategy is believed to induce indirect hypoxia and stimulate adaptive and protective responses. RIPC was tested in adults undergoing cardiovascular surgery [165] and in pediatric populations undergoing cardiopulmonary bypass [166]. However, neither study reported significant myocardial protection compared to the control group. Only the pediatric trial reported a reduction in acute kidney injury, a common complication following cardiac surgery.

In contrast to the intentional, controlled hypoxia induced by AIH or RIPC, clinical scenarios such as OSA involve pathological chronic intermittent hypoxia (CIH). It is associated with increased cardiovascular risk and hypertension. Although data on the regenerative effects of CIH in OSA are limited, a recent review concludes that a sustained hypoxic profile impairs wound healing in both surgical and diabetic wounds [167]. Moreover, an additional perspective arises from the interaction between AIH and OSA-related hypoxia. A recent study showed that patients with arterial hypertension and controlled OSA, who were treated with AIH in cycles of 15 min of moderate hypoxia followed by 2 min of normoxia, showed significant improvements in their systolic blood pressure levels after two weeks. These results suggest that, even in patients with OSA, controlled moderate AIH can provide benefits [168].

The studies discussed represent a significant step forward in understanding the role of hypoxia in tissue protection and regeneration. However, standardized clinical protocols are required to enable cross-study comparisons that allow predictable and consistent outcomes. Therefore, it is critical to consider population variability and individual responses, as these factors may profoundly influence therapeutic outcomes. Furthermore, future research in this field needs to be expanded, utilizing precise protocols and larger, more representative patient cohorts. Finally, it is crucial not to overlook the potential risks of using hypoxia as a therapeutic tool. These include sustained inflammation, endothelial dysfunction, mitochondrial impairment, and fibrosis, among others. Such hazards have been detailed in a recent review, calling for caution and thorough evaluation before translating hypoxia-based approaches into clinical practice [169].

## 6. Conclusions, Challenges, and Perspectives

Several genes, proteins, and signaling pathways emerge as common mediators of regenerative responses under hypoxia. At the center of this response is the action of HIF, which regulates angiogenesis (via VEGF), metabolic reprogramming (toward glycolysis), stem cell maintenance, and immune modulation (via NF-κB, NRF2, and STAT3) [15,50]. Moreover, key transcriptional regulators, such as NOTCH, Wnt (especially the non-canonical branch), TGF-β/Smad2/3, and several microRNAs, participate in coordinating cellular proliferation, differentiation, and the control of fibrosis in hypoxic environments [106,149]. Although tissue-specific differences exist, these shared regulators suggest that hypoxia triggers a conserved regenerative program across biological systems.

The overall evidence reviewed highlights both the regenerative potential and the risks associated with hypoxia. While preconditioned, short, or moderate hypoxic exposures can stimulate repair through angiogenesis, enhanced stemness, and metabolic adaptation, sustained or severe hypoxia may instead promote fibrosis, cellular exhaustion, or maladaptive remodeling [97]. These contrasting outcomes command the urgent need for future studies. Thus, it is essential to apply standardized experimental conditions, including oxygen concentration and administration, duration, and cellular context, to ensure reproducibility among studies that facilitates meaningful comparisons and minimizes conflicting interpretations [99].

In addition, experimental findings are influenced not only by methodological variability but also by intrinsic biological differences in hypoxia tolerance, even among genetically similar or inbred populations. For example, consistent interindividual differences in hypoxia tolerance have been reported in the Gulf killifish (*Fundulus grandis*), demonstrating that such variability can be repeatable even within the same species [170]. Similarly, (*Rattus norvegicus*) (Wistar strain) displays distinct subpopulations with high, medium, and low resistance to acute severe hypoxia, with survival differences reaching up to 90% under comparable conditions [171]. Other studies have shown that this variability also occurs between strains, particularly between August rats (*Rattus norvegicus*), (high resistance) and Wistar rats (low resistance), which is associated with specific mitochondrial mechanisms [70]. Together with the impact of technical factors on oxygen delivery and exposure conditions [157,172,173], and the recommendations to account for baseline physiological variability [174], these findings emphasize the need to assess and report such variations, and to clearly define and trace experimental models to strengthen the robustness and reproducibility of hypoxia research.

Building on this, comparative research using highly regenerative model organisms, such as zebrafish (*Danio rerio*) or amphibians (*Xenopus* sp.), offers unique opportunities to address some of these challenges. These models not only allow controlled analyses of hypoxia tolerance variability but also help identify genetic, epigenetic, and molecular traits that are absent in mammals. Understanding how such species activate developmental programs through HIF-mediated signaling (e.g., Isl1, Nkx2.5, Wnt pathways) may help elucidate why mammals fail to regenerate efficiently [82,84,143]. These insights could not only inform regenerative medicine but also reveal evolutionarily conserved mechanisms that may be reactivated or modulated pharmacologically to enhance tissue repair in humans.

Finally, the translation of preclinical findings into clinical settings faces similar challenges. Clinical trials investigating acute intermittent hypoxia protocols in humans have shown promising effects on neuroregeneration and cardiovascular function. However, they also reveal marked interindividual variability and inconsistencies in oxygen dosing, exposure duration, and patient baseline conditions. These limitations must be addressed by standardized, well-controlled, and larger-scale clinical studies that safely and effectively exploit hypoxia-based therapies. Advances in this field will depend on bridging preclinical and clinical research, reducing methodological variability, and carefully defining thresholds for beneficial versus harmful hypoxic exposures that translate fundamental knowledge into safe and effective therapeutic protocols.

These findings convey a final message about the power of hypoxia, representing both a significant biological challenge and a promising opportunity for innovation in regenerative therapies.

## Figures and Tables

**Figure 1 ijms-26-09272-f001:**
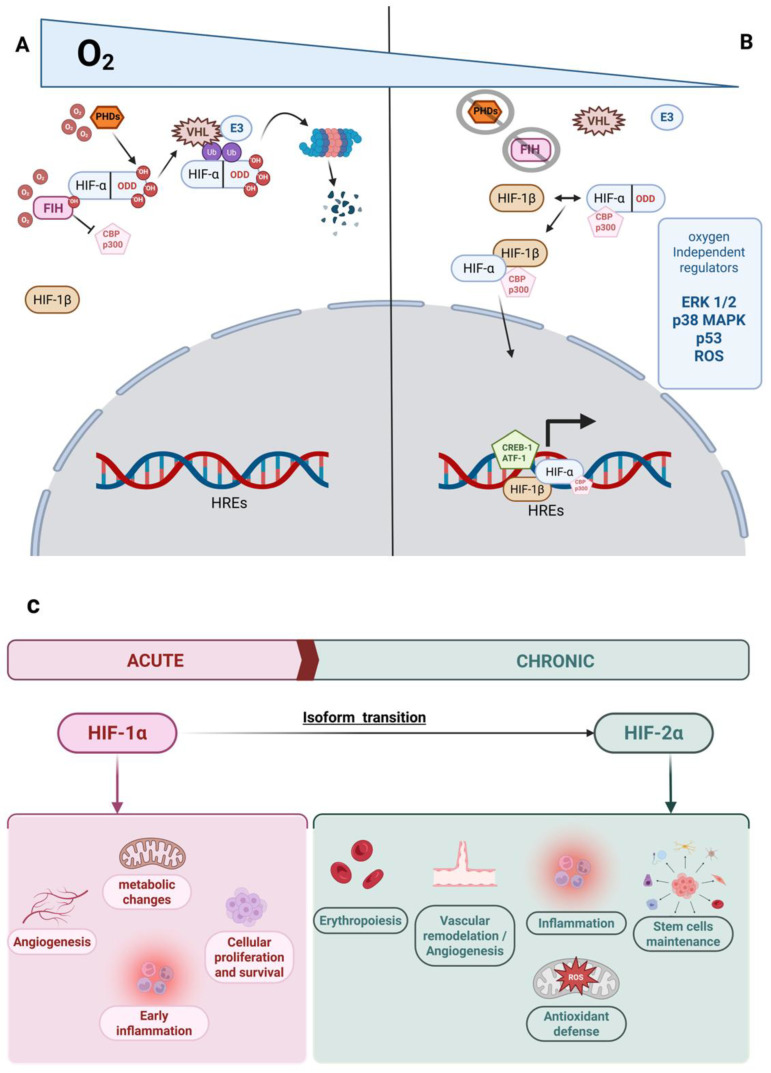
Oxygen-dependent regulation and isoform-specific functions of HIF-α under hypoxia. (**Panel A**) Under normoxic conditions, HIF-α is hydroxylated by prolyl hydroxylases (PHDs), which enables its recognition by the von Hippel–Lindau (VHL) protein. Together with elongins, VHL forms part of an E3 ubiquitin ligase complex that mediates the ubiquitination of HIF-α and its subsequent proteasomal degradation. Additionally, oxygen availability allows factor inhibiting HIF (FIH) to hydroxylate an asparagine residue in HIF-α, which prevents its interaction with transcriptional coactivators CBP and p300, thereby inhibiting its transcriptional activity. (**Panel B**) Under hypoxic conditions, PHD activity is suppressed, allowing HIF-α to stabilize, translocate to the nucleus, dimerize with HIF-β, and bind to hypoxia response elements (HREs), promoting the transcription of target genes. This panel also includes oxygen-independent modulators of HIF-1α. Positive regulators, such as ERK1/2, p38 MAPK, ATF-1, and CREB-1, enhance HIF-1α stability or transcriptional activity, whereas p53 acts as a negative regulator by promoting its degradation [61]. Although not shown graphically, epigenetic mechanisms such as CpG methylation and microRNAs are discussed in the text as contributors to isoform-specific regulation. For instance, CpG methylation in the EPO gene modulates its accessibility to HIF-2α, and miRNAs, such as miR-18a, selectively inhibit HIF-1α [60,61]. (**Panel C**): This section compares the roles of HIF-1α and HIF-2α in acute and chronic hypoxia, respectively. HIF-1α drives early adaptive responses, including glycolysis, angiogenesis, initial inflammatory signaling, and cell proliferation or survival. HIF-2α predominates in prolonged hypoxia, supporting erythropoiesis, extracellular matrix remodeling, antioxidant responses, and the maintenance and self-renewal of stem cells [61]. Although intermittent hypoxia is not explicitly illustrated, it is acknowledged to promote dynamic modulation between both isoforms, with a predominant role for HIF-1α [61]. Together, these mechanisms enable oxygen-dependent, context-specific cellular adaptation.

**Figure 2 ijms-26-09272-f002:**
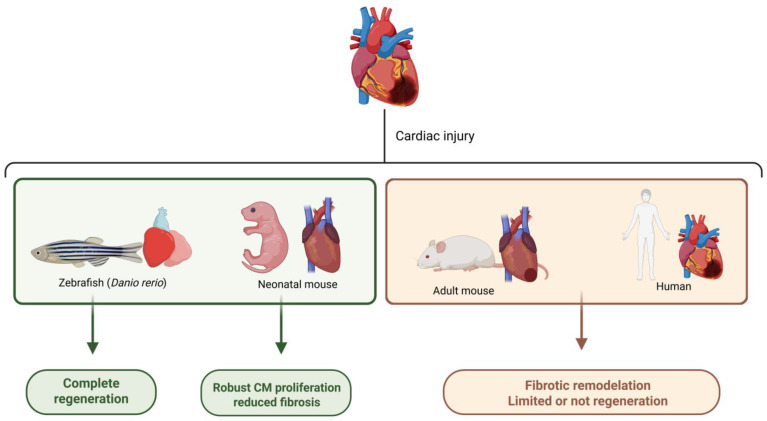
Comparative cardiac regenerative capacity across species. Following cardiac injury, zebrafish (*Danio rerio*) exhibit complete regeneration, while neonatal mice show robust cardiomyocyte (CM) proliferation with reduced fibrosis. In contrast, adult mice and humans mainly undergo fibrotic remodeling, with limited or absent regenerative responses.

**Figure 3 ijms-26-09272-f003:**
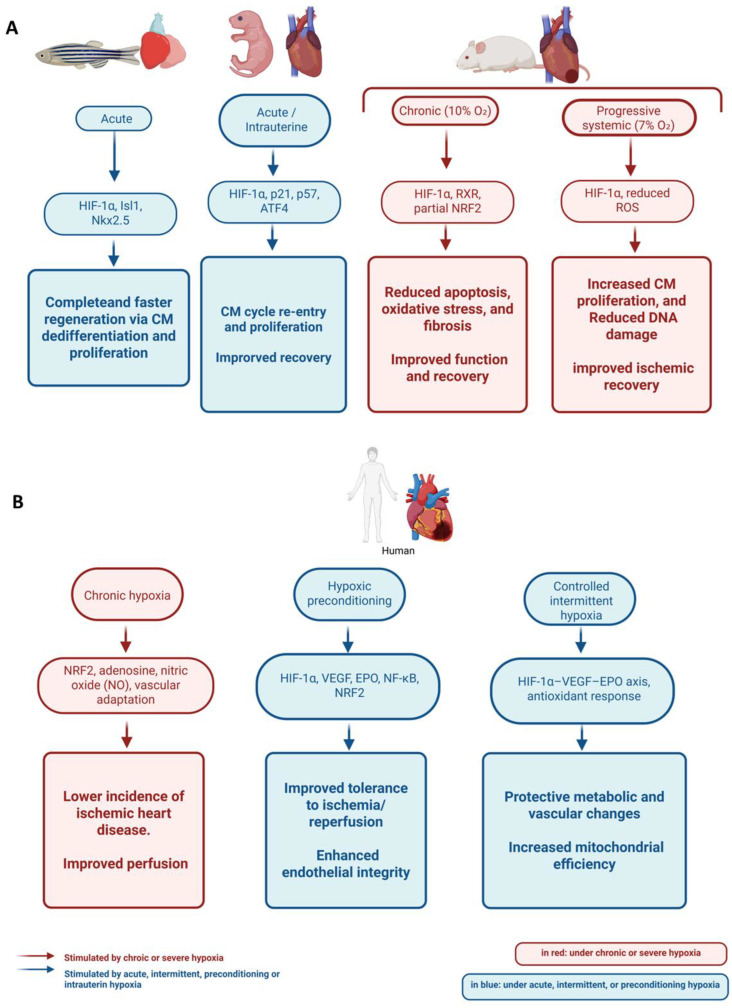
Hypoxia-Induced Cardiac Responses: Regeneration in Experimental Models and Cardioprotection in Humans. (**A**) illustrates the regenerative capacity in zebrafish (*Danio rerio*) and neonatal mice. In zebrafish, acute hypoxia triggers cardiomyocyte (CM) proliferation, dedifferentiation, and angiogenesis via hypoxia-inducible factor 1 alpha (HIF-1α), ISL LIM homeobox 1 (Isl1), and NK2 homeobox 5 (Nkx2.5). In neonatal mice, HIF-1α suppresses cell cycle inhibitors, such as p21, p57, and activating transcription factor 4 (ATF4), thereby promoting CM cell-cycle reentry during the early postnatal period. (**B**) summarizes human cardioprotective adaptations. In high-altitude populations, long-term hypoxia is associated with a reduced cardiovascular risk, mediated by NRF2 activation, nitric oxide (NO) signaling, and adenosine release. In hypoxic preconditioning, transient low oxygen levels enhance ischemia tolerance via HIF-1α, erythropoietin (EPO), nuclear factor kappa B (NF-κB), and NRF2 activation. Controlled intermittent hypoxia stimulates the HIF-1α–VEGF–EPO axis and antioxidant responses, improving metabolic and vascular function as well as mitochondrial efficiency. Collectively, these findings highlight the context-dependent role of hypoxia in either promoting cardiac regeneration or conferring cardioprotection.

**Figure 4 ijms-26-09272-f004:**
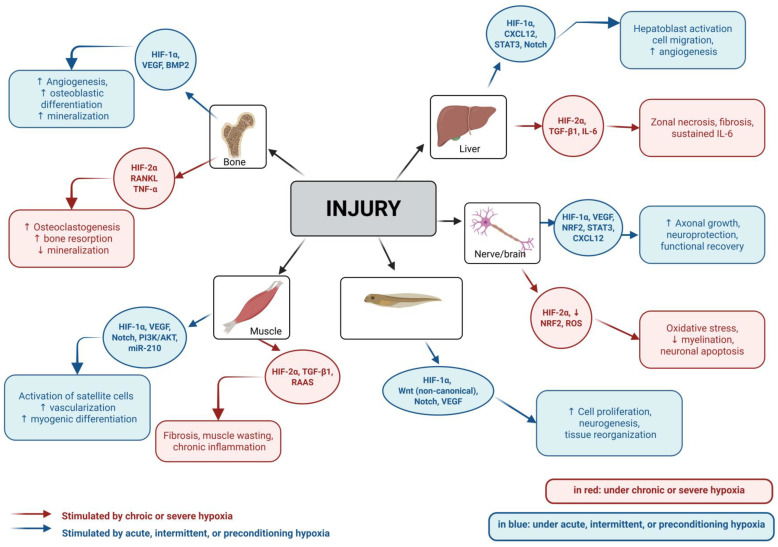
Context-specific effects of hypoxia types on regeneration across multiple tissues. This figure summarizes the differential effects of acute, intermittent, preconditioning, and chronic hypoxia on tissue regeneration. Beneficial responses (blue) include increased vascularization, progenitor activation, metabolic shifts, and tissue remodeling, mediated by HIF-1α, VEGF, Notch, Wnt, PI3K/AKT, STAT3, CXCL12, and NRF2. In structures such as tail, studied in *(Xenopus laevis*), acute hypoxia promotes early ROS production and HIF-1α stabilization, improving blastema formation, inflammation, and cellular activation. Later stages involve sustained HIF-2α expression, supporting angiogenesis and differentiation. Hypoxia also enhances the expression of genes like *notch1*, *msx1*, *msx2*, *cdx4*, and *fgf10/20*, relevant to regeneration. In muscle, acute or intermittent hypoxia supports satellite cell renewal, angiogenesis, and functional recovery, while chronic exposure impairs differentiation and promotes fibrosis. In bone, hypoxia can enhance osteogenesis and vascularization, though under severe or chronic conditions, it reduces mineralization and promotes osteoclast activity. Liver regeneration benefits from VEGF-mediated neovascularization and transient inflammatory activation under hypoxia. In neural tissues, hypoxia-preconditioned cells and extracellular vesicles improve regeneration, axonal growth, and post-ischemic repair. Maladaptive responses (red) include mitochondrial dysfunction, persistent inflammation, fibrosis, and a reduced capacity for repair. These are driven by HIF-2α, TGF-β1, IL-6, and reduced NRF2 activity. In bone, chronic hypoxia and elevated EPO may impair perfusion and promote resorption. Color coding indicates the type of hypoxia: blue arrows denote beneficial responses to acute, intermittent, or preconditioning hypoxia, while red arrows denote the maladaptive effects of chronic hypoxia. These context-specific outcomes demonstrate that hypoxia is neither inherently positive nor negative, but rather exerts tissue- and duration-dependent effects. Collectively, these findings demonstrate how hypoxia affects regenerative processes across various tissues, including cardiac, muscular, and bone repair, as well as vascular remodeling and the regeneration of complex structures. Building on these mechanistic insights, the next section, Section 5, explores how hypoxia is being translated into regenerative medicine, covering preconditioning strategies for stem cells, bioengineering approaches such as extracellular vesicles and hydrogels, and emerging clinical applications designed to enhance tissue repair while emphasizing the need for methodological standardization due to potential risks associated with hypoxic exposure.

**Table 2 ijms-26-09272-t002:** Regulatory mechanisms underlying the functional shift from HIF-1α to HIF-2α.

Hypoxia Type	Regulator	Target Isoform	Effect	Ref.
**Acute**	Sirtuin 1 (SIRT1)	HIF-1α	Deacetylates HIF-1α, enhancing its transcriptional activity on target genes	[60]
	Heat shock protein 90 (HSP90)	HIF-1α	Stabilizes and enhances HIF-1α function	[60]
	PHD3 stimulated by HIF-1α	HIF-2α	Preferentially increases HIF-2α degradation	[64]
**Chronic**	NF-κB essential modulator (NEMO)	HIF-2α	Increases HIF-2α stability and availability (via STAT3 coactivation)	[63]
	Ets-1 transcription factor (E-twenty-six oncogene)	HIF-2α	Acts as a coactivator	[63]
	HAF/SART1 (Hypoxia-associated factor)	HIF-1α	Promotes proteasomal degradation	[63]
		HIF-2α	Stimulates HIF-2α activity	[60,63,64]
	aHIF (antisense transcript of HIF-1α)	HIF-1α	Binds HIF-1α mRNA and promotes its degradation	[63]
	FIH-1 (Factor Inhibiting HIF-1α)	HIF-1α	Hydroxylates HIF-1α and blocks cofactor binding	[60]
	Hsp70/CHIP (Heat shock protein 70/C-terminus of Hsc70-interacting protein)	HIF-1α	Ubiquitination and degradation	[60]
	RACK1 (Receptor for Activated C Kinase 1)	HIF-1α	Proteasomal degradation/inhibits HSP90	[60]
	KLF2 (Krüppel-like factor 2)	HIF-1α	Inhibits HIF-1α interaction with HSP90	[60]
	Reduced stability of HIF-1α mRNA	HIF-2α and HIF-3α	Predominance of HIF-2α and HIF-3α isoforms	[60]

Table 2 summarizes some key molecular regulators involved in the functional transition from HIF-1α to HIF-2α under different hypoxic conditions. Acute hypoxia stabilizes HIF-1α, whereas chronic hypoxia favors the predominance of HIF-2α through mechanisms that enhance its stability or promote HIF-1α degradation. Regulatory factors include chaperones, transcription factors, ubiquitin ligases, and post-translational modulators. These processes enable a context-dependent adjustment of HIF isoform activity. References correspond to numbered citations in the main text.

## Data Availability

Not applicable.

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
