# Peer review of "Hypoxia and Tissue Regeneration: Adaptive Mechanisms and Therapeutic Opportunities"

_ijms, 2025, doi:10.3390/ijms26199272_

Round 1

Reviewer 1 Report

Comments and Suggestions for Authors

The review article by Velez et al provides a comprehensive summary which highlights the effects of reduced oxygen availability on regeneration both in vitro and in various animal models. The review incorporates both mechanistic and genomic insight into the effects of long term, short term and intermittent hypoxia in multiple species. In addition, the review summarizes both benefits and detrimental effects of hypoxia on key organ systems.
General Comments:
1. The description of benefits of controlled intermittent hypoxia in humans should be elaborated upon and described more clearly.
2.Add references to the statements on lines 255 and 256.
3. Provide further clarification on line 255-256 stating “inflammation response is also regulated to enhance tissue repair by promoting proliferation.”
4.The resolution of the Figures, in particular Figure 3 parts B and C, should be increased to improve readability.
5.Section 4.3 Other Tissues and Complex Structures seems out of place here.
6. Section 5 entitled Cardiac Regeneration Under Hypoxia should be moved up to become Section 4.3.
7.The information provided in section 4.3 entitled “Other Tissues and Complex Structures” is a mix of different situations in which hypoxia may benefit regenerative outcomes. This section should be eliminated entirely. As currently presented, it detracts from the flow of the manuscript.
8. A separate section focused entirely on the impact of short, interim and long-term hypoxia on isolated stem cells as a pre-conditioning treatment prior to administration of the stem cells for studies focused on tissue regeneration would be beneficial to include in this manuscript.
9. The conclusions are consistent with the summary of evidence published to date regarding the adaptive mechanisms that have been reported following exposure to reduced oxygen. The review highlights the common transcriptomic and signaling pathway changes induced by hypoxia and the potential for beneficial effects on regeneration yet includes the potential risks of long- term exposure to hypoxia.
10.The references are appropriate.
Minor Edits:
Line 229 remove the s from population
Line 234 Remove the extra space between at and this
Line 236 change revealing to reveal
Line 250-252 reword the sentence
Line 256 replace instants with instance
Line 425 remove the word informed
Line 605 remove the word While
Line 848 reword this sentence

Comments on the Quality of English Language

Suggested minor edits are included in the review file. 

Author Response

Dear Reviwer:

We want to thank you for your time and willingness to review our work thoroughly. Each of your observations was highly valuable and allowed us to look at the manuscript from different perspective, thereby strengthening it. In response to the concept of the reviewers´ and to enhance the flow and coverage of information presented in our manuscript, we restructured the content outlined it as follows: 

1.Introduction 

2.Hypoxia Types and Systemic Adaptation 

  1. Cellular Mechanisms under Hypoxia 

3.1 Cellular mechanisms of adaptation to hypoxia mediated by HIF 

3.2 Non-HIF-dependent mechanisms of adaptation to hypoxia 

  1. Hypoxia in Regenerative Processes

4.1 Cardiac Regeneration 

4.2 Muscle regeneration 

4.3 Bone Regeneration 

4.4 Hypoxia and vascular responses during regeneration 

4.5 Hypoxia in appendages regeneration 

  1. Hypoxia in regenerative medicine

5.1 Stem cells and bioengineering applications 

5.1.1 Heart 

5.1.2 Muscle 

5.1.3 Bone 

5.1.4 Neuro 

5.2 Clinical applications 

  1. Conclusions, Challenges, and Perspectives 

Our point by point response is as follows: 

Comments 1: The description of benefits of controlled intermittent hypoxia in humans should be elaborated upon and described more clearly. 

Response 1: The description of the beneficial effects of controlled intermittent hypoxia in humans has been clarified in Section 2 (lines 307–315). We also contrasted with the adverse effects of chronic intermittent hypoxia (lines 326–337). Additionally, emphasis was placed on this topic, and further details were added in Section 5.2 on clinical applications. 

Comments 2 and 3: 

- Add references to the statements on lines 255 and 256. 

- Provide further clarification on lines 255-256 stating “inflammation response is also regulated to enhance tissue repair by promoting proliferation.” 

Response 2 and 3: The sentence “inflammation response is also regulated to enhance tissue repair by promoting proliferation” has been clarified and is now properly supported by the requested reference, currently located in (lines 312–316) of the revised manuscript.  

Comments 4: The resolution of the Figures, in particular Figure 3 parts B and C, should be increased to improve readability. 

Response 4: The images were included within the manuscript using the journal’s provided template, which may have affected their resolution. However, to improve image quality, they have now been saved at a higher resolution (600 DPI, extra high resolution) and will be uploaded as separate JPG files and inserted at a larger size. Figure 3 due to its high amount of graphic content and text has reduced readability; therefore, we decided to split it into two separate figures, now Figures 2 and 3. We consider this change has enhanced the overall resolution and readability.   

Comments 5: Section 4.3 Other Tissues and Complex Structures seems out of place here. 

Response 5: We understand your observation and see your point. Thus, considering that the vascular contribution is essential to the general regeneration process, and that the regenerative process in appendages is valuable for its own complexity, we have restructured the content of this section to provide greater clarity as shown in the outlined of updated content. We consider this contextualize better those topics and provide a more coherent and fluid narrative.  

Comments 6: Section 5, entitled Cardiac Regeneration Under Hypoxia should be moved up to become Section 4.3. 

Response 6: In response to your suggestion, Section 4 was reorganized to include the heart. The section originally numbered 4.3, as mentioned in the previous comment, has also been reorganized, retaining the original information with minor refinements. 

Comments 7: The information provided in section 4.3 entitled “Other Tissues and Complex Structures” is a mix of different situations in which hypoxia may benefit regenerative outcomes. This section should be eliminated entirely. As currently presented, it detracts from the flow of the manuscript. 

Response 7: Given contradictory appreciation of this section with reviewer 2. We address this comment reorganizing the content as outlined initially and as described previously in the response to Comment 5.  

Comments 8: A separate section focused entirely on the impact of short, interim, and long-term hypoxia on isolated stem cells as a preconditioning treatment prior to administration of the stem cells for studies focused on tissue regeneration would be beneficial to include in this manuscript. 

Response 8: In the original manuscript, hypoxic preconditioning of stem cells, the release of extracellular vesicles derived from these cells, and other approaches mainly based on in vitro techniques were mentioned but scattered throughout the text. This topic has now been consolidated and revised in its own subsection in 5.1. 

Comments 9: The conclusions are consistent with the summary of evidence published to date regarding the adaptive mechanisms that have been reported following exposure to reduced oxygen. The review highlights the common transcriptomic and signaling pathway changes induced by hypoxia and the potential for beneficial effects on regeneration, yet includes the potential risks of long-term exposure to hypoxia. 

Response 9: We appreciate your comment. In the revised version of our manuscript, and considering also the request of reviewer 2, we have expanded the conclusions, placing greater emphasis on the current need to standardize protocols and guidelines when working with hypoxia. We also added relevant aspects related to plausible biological variability and the potential risks associated with the application of hypoxia-based therapies. 

Comments 10: The references are appropriate. Minor edits and English improvement. 

Response 10: A more detailed revision of the language quality has been made. We have applied improvements to the English writing hoping it results in greater clarity and a smoother flow for readers throughout the manuscript.  

We greatly appreciate your comments and suggestions. We have found them invaluable to enhance the impact and overall perspective of our manuscript so that we have addressed them all to our best. We hope you find them appropriate as we intend them to be. 

Reviewer 2 Report

Comments and Suggestions for Authors

The article by Velez and colleagues addresses an important and rapidly evolving field of research on the role of hypoxia in tissue regeneration. The topic holds significant potential for clinical applications, particularly in regenerative medicine. A valuable aspect of the work, in my view, is the authors comprehensive analysis of hypoxia effects on various tissues (bone, muscle, nervous, and cardiac). The article is well-organized, with a logical division into sections, including an introduction, types of hypoxia, cellular mechanisms, regenerative processes, and conclusions. The inclusion of figures and tables (a comparative table of genomic adaptations) enhances the comprehension of complex information. I have the following recommendations to further improve the manuscript:

1. While the article provides a detailed discussion of mechanisms and animal experiments, clinical studies are rarely mentioned. It would be beneficial to include more data on the application of hypoxic approaches in medicine, including ongoing clinical trials.

2. The effects of chronic hypoxia on the body (fibrosis, maladaptation) are only briefly addressed. Expanding this section, particularly in the context of potential risks for therapeutic applications, would be worthwhile.

3. The focus is primarily on HIF-dependent pathways, while other mechanisms (the role of mitochondria or non-HIF transcriptional factors) receive less attention. Elaborating on these aspects, particularly by incorporating information on mitochondrial proteome remodeling (including changes in mitochondrial calcium uniporter (MCU) levels) and the mitochondrial permeability transition pore, would significantly strengthen the article. For example, it is well-established that parameters of Ca²⁺ uptake via MCU and its release through MPT pore opening differ markedly between rats with high and low resistance to acute hypoxia. Other known adaptations should also be discussed.

4. The authors note that research findings are contradictory due to variations in experimental conditions. This is an important observation, and it should be further emphasized that even within inbred animal models (like the widely used Wistar rat strain), there are individuals with high and low resistance to acute hypoxia. Researchers often overlook this factor, which can significantly influence study outcomes.

The authors state that «The authors used ChatGPT (OpenAI, 2025) for language support to improve grammar and document readability». While I have no major objections to this, I believe such tools should be used with caution. I leave this matter to the editor discretion.

Author Response

Dear Reviwer:

We want to thank you for your time and willingness to review our work thoroughly. Each of your observations was highly valuable and allowed us to look at the manuscript from different perspective, thereby strengthening it. In response to the concept of the reviewers´ and to enhance the flow and coverage of information presented in our manuscript, we restructured the content outlined it as follows: 

1.Introduction 

2.Hypoxia Types and Systemic Adaptation 

  1. Cellular Mechanisms under Hypoxia 

3.1 Cellular mechanisms of adaptation to hypoxia mediated by HIF 

3.2 Non-HIF-dependent mechanisms of adaptation to hypoxia 

  1. Hypoxia in Regenerative Processes

4.1 Cardiac Regeneration 

4.2 Muscle regeneration 

4.3 Bone Regeneration 

4.4 Hypoxia and vascular responses during regeneration 

4.5 Hypoxia in appendages regeneration 

  1. Hypoxia in regenerative medicine

5.1 Stem cells and bioengineering applications 

5.1.1 Heart 

5.1.2 Muscle 

5.1.3 Bone 

5.1.4 Neuro 

5.2 Clinical applications 

  1. Conclusions, Challenges, and Perspectives

Our point by point response is as follows: 

Comments 1: While the article provides a detailed discussion of mechanisms and animal experiments, clinical studies are rarely mentioned. It would be beneficial to include more data on the application of hypoxic approaches in medicine, including ongoing clinical trials. 

Response 1: We agree on the importance of including clinical advances that evaluate the effects of hypoxia in the context of both tissue regeneration and tissue protection, as well as its potential risks. To address this point, a new subsection 5.2 (Clinical applications) has been incorporated as outlined above. 

Comments 2: The effects of chronic hypoxia on the body (fibrosis, maladaptation) are only briefly addressed. Expanding this section, particularly in the context of potential risks for therapeutic applications, would be worthwhile. 

Response 2: We emphasized this aspect in Section 2 (lines 263–290), specifically regarding chronic hypoxia, and in lines 326–337 for intermittent hypoxia, particularly chronic exposure. We also sought to better balance the adverse effects with the potentially beneficial ones throughout the manuscript. 

Comments 3: The focus is primarily on HIF-dependent pathways, while other mechanisms (the role of mitochondria or non-HIF transcriptional factors) receive less attention. Elaborating on these aspects, particularly by incorporating information on mitochondrial proteome remodeling (including changes in mitochondrial calcium uniporter (MCU) levels) and the mitochondrial permeability transition pore, would significantly strengthen the article. For example, it is well-established that parameters of Ca² uptake via MCU and its release through MPT pore opening differ markedly between rats with high and low resistance to acute hypoxia. Other known adaptations should also be discussed. 

Response 3: We overlooked the importance of expanding this aspect in our manuscript. As a result, Section 3 has now been divided into two subsections: 3.1 HIF-dependent mechanisms and 3.2 non-HIF-dependent mechanisms of adaptation to hypoxia. The mechanisms considered most relevant for subsection 3.2 have been included and can be reviewed starting from line 515 of the manuscript. 

Comments 4: The authors note that research findings are contradictory due to variations in experimental conditions. This is an important observation, and it should be further emphasized that even within inbred animal models (like the widely used Wistar rat strain), there are individuals with high and low resistance to acute hypoxia. Researchers often overlook this factor, which can significantly influence study outcomes. 

Response 4: We have expanded and stressed in the discussion on standardizing methodologies and experimental protocols used to assess hypoxia, as well as the variability observed among individuals, strains, and populations. We emphasized and further addressed these points in Section 6 (Conclusions, Challenges, and Perspectives). 

Comments 5: The authors state that «The authors used ChatGPT (OpenAI, 2025) for language support to improve grammar and document readability». While I have no major objections to this, I believe such tools should be used with caution. I leave this matter to the editor’s discretion. 

Response 5: We completely understand your concerns and agree that this tool should be used with caution and clear limits. We used it with caution and reported its usage during the submission process, as specifically instructed by the journal. We want to reiterate that the use of ChatGPT was solely intended to support linguistic editing, as English is not our first language. The full content of the manuscript is entirely the result of our careful review of original articles, and the structure was generated based on our own need to put all the available data together. 

We greatly appreciate your comments and suggestions. We have found them invaluable to enhance the impact and overall perspective of our manuscript so that we have addressed them all to our best. We hope you find them appropriate as we intend them to be. 

Round 2

Reviewer 2 Report

Comments and Suggestions for Authors

The authors responded to all my concerns and greatly improved the presentation of the manuscript.